


# A multi-method autonomous assessment of primary productivity and export efficiency in the springtime North Atlantic

Nathan Briggs[1], Kristinn Guðmundsson[2], Ivona Cetinić[3], Eric D'Asaro[4], Eric Rehm[5], Craig Lee[4], Mary Jane Perry[6]

[1]National Oceanography Centre, Southampton SO14 3ZH, UK
[2]Marine Research Institute, PO Box 1390, 121 Reykjavík, Iceland
[3]Ocean Ecology Laboratory, NASA Goddard Space Flight Center Code 616, Greenbelt, MD 20771, USA
[4]Applied Physics Laboratory and School of Oceanography, University of Washington, Seattle, WA 98105, USA
[5]Département de Biologie et Québec-Océan, Université Laval, Québec, QC G1V 0A6, Québec, Canada
[6]Darling Marine Center, School of Marine Sciences, University of Maine, Walpole, ME 04573, USA

*Correspondence to*: Nathan Briggs (natebriggs@gmail.com)

## Abstract

Fixation of organic carbon by phytoplankton is the foundation of nearly all open-ocean ecosystems and a critical part of the global carbon cycle. But quantification and validation of ocean primary productivity at large scale remains a major challenge, due to limited coverage of ship-based measurements and the difficulty of validating diverse measurement techniques. Accurate primary productivity measurements from autonomous platforms would be highly desirable, due to much greater potential coverage. In pursuit of this goal we estimate gross primary productivity over two months in the springtime North Atlantic from an autonomous Lagrangian float using diel cycles of particulate organic carbon derived from optical beam attenuation. We test method precision and accuracy by comparison against entirely independent estimates from a locally parameterized model based on chlorophyll *a* and light measurements from the same float. During nutrient replete conditions (80% of the study period), we obtain strong relative agreement between the independent methods across an order of magnitude of productivities ($r^2$=0.97), with slight under-estimation by the diel cycles method (-19±5 %). At the end of the diatom bloom, this relative difference increases to -58 % for a six-day period, likely a response to $SiO_4$ limitation, which is not included in the model. In addition, we estimate gross oxygen productivity from $O_2$ diel cycles and find strong correlation with diel cycles-based gross primary productivity over the entire deployment, providing further qualitative support to both methods. Finally, simultaneous estimates of net community productivity, carbon export and particle size suggest that bloom growth is halted by a combination of reduced productivity due to $SiO_4$ limitation and increased export efficiency due to rapid aggregation. After the diatom bloom, high chlorophyll *a* normalized productivity indicates that low net growth during this period is due to increased heterotrophic respiration and not nutrient limitation. These findings represent a significant advance in the accuracy and completeness of upper ocean carbon cycle measurements from an autonomous platform.



# 1 Introduction

Understanding of ocean primary productivity (PP), the origin of nearly all organic carbon available to marine organisms, is critical to understanding marine ecosystems and predicting how they might respond to human activities. Because human influences such as climate change and fishing have global impact, improvements in global, mechanistic understanding of the

drivers of PP and also the effects of PP on ecosystems and their services should be of great value. However, global understanding is limited by the difficulty of measuring PP, which traditionally involves incubation experiments and/or radio- or stable isotope analysis, requiring cost, expertise, and ship sampling. Understanding is further limited by the difficulty in validating PP, as each method has potential sources of bias, but generally no two methods measure the exact same quantity at the same temporal scale. Therefore, it is often unclear whether discrepancies between independent measurements are caused

by biases or real differences. Satellite PP algorithms and global models can achieve the desired coverage, but these products still must be validated, ideally using an in situ dataset of confirmed accuracy that spans many years, in all seasons and in all oceans. Autonomous platforms can achieve such in situ coverage at a fraction of the cost of ship-based sampling, so the ability to estimate PP from an autonomous platform and validate these estimates using independent methods is highly desirable, both for directly enhancing understanding of ocean ecosystems and validating the models and satellite products

that can approach true continuous global coverage.

Methods for estimating PP from diel cycles in particulate beam attenuation $c_p$ (Siegel et al. 1989; Claustre et al. 1999; Cullen et al. 1992; Kinkade et al. 1999; Dall'Olmo et al. 2011; Gernez et al. 2011; Omand et al. 2017) or $O_2$ (Caffrey 2003; Hamme et al. 2012; Nicholson et al. 2015) are suited for application to autonomous platforms, many of which already carry $O_2$

sensors and/or transmissometers. These methods rely on the light-dependence of PP, which causes a diel cycle in $O_2$ and in $c_p$ (due to its correlation with particulate organic carbon, POC). However, other factors such as zooplankton vertical migrations, mixing events, $O_2$ airsea flux, and POC/$c_p$ ratios may have diel cycles that introduce bias in these PP estimates, so they cannot be relied upon without validation. Comparisons so far between diel cycles and independent PP estimates have been encouraging, generally agreeing within a factor of two to three (Cullen et al. 1992; Walsh et al. 1995; Kinkade et al.

1999; Hamme et al. 2012; Nicholson et al. 2015), but the independent estimates have not been of the same quantity at the same temporal scale, so these comparisons do not provide strong constraints on the accuracy of this method.

In this study we take three significant steps towards the goal of enhancing our understanding of ocean ecosystems by increasing coverage of accurate in situ PP estimates using autonomous platforms. First, we use diel cycles in measurements

of $c_p$ and $O_2$ to simultaneously estimate two related quantities, gross primary productivity of particulate organic carbon (GPP) and gross oxygen productivity (GOP), in the surface mixed layer over a two-month period from an autonomous Lagrangian float. Two our knowledge, this is the first time GPP and GOP have been simultaneously calculated using diel cycles from any platform, let alone autonomously. Second, we compare our diel cycles-based GPP estimates with entirely





independent estimates of the same quantity at the same spatial and temporal scale across a wide dynamic range of productivities. Again, to our knowledge, this represents the most rigorous validation of the diel cycles method to date. Third, we apply our mixed layer PP estimates, in conjunction with mixed layer $O_2$, $NO_3$, and POC budgets, to better understand how PP, heterotrophic respiration, and sinking flux all interact to regulate mixed-layer biomass in our study system: the

spring diatom bloom in the Iceland Basin.

## 2 Methods

### 2.1 Study area and Platforms

The data presented here were collected by an autonomous Lagrangian mixed-layer float, two ships, and three autonomous Seagliders during the North Atlantic Bloom 2008 (NAB08) project. All data used here are available online at

http://www.bco-dmo.org/project/2098. The float was deployed on April 4 in the Iceland Basin at 59°N, 20.5°W, near the 60°N site of the 1989 Joint Global Ocean Flux Study (JGOFS). The NAB08 project centered around the float, which was designed to drift in the surface mixed layer, mimicking the movement of plankton, except for daily profiles to 250 m (D'Asaro 2003). The float gathered data for two months, drifting northwest towards the Reykjanes Ridge, and ceased collecting data on May 25 at 61.8°N, 26.7°W (Fig. 1; black line), and was recovered on June 3. The timing of the daily float

profiles was irregular until April 14, after which the float profiled each day between 15:00 GMT and dusk. The float carried an array of sensors, including two SBE-43-CTs for temperature and salinity, a WET Labs C-Star transmissometer for particulate organic carbon (POC), via particulate beam attenuation $c_p$, a WET Labs FLNTU (fluorescence and turbidity meter) for chlorophyll a fluorescence and POC, via particulate optical backscattering $b_{bp}$, a Seabird SBE-43 and an Aanderaa optode for oxygen, an ISUS (In Situ Ultraviolet Spectrophotometer) for $NO_3$, a LICOR LI-192SA for planar

photosynthetically active radiation (PAR). See Table 1 for a list of abbreviations used in more than one subsection. Three cruises provided calibration data for the float's sensors as well as more detailed biological and chemical measurements: a deployment cruise by the R.V. *Bjarni Saemundsson* (April 3-5), a process cruise by the R.V. *Knorr* (May 2-21) and a float "rescue" cruise by the R.V. *Bjarni Saemundsson* (June 4-5). The ships collected both in situ measurements and discrete water samples via an overboard CTD package, which profiled to 600 m. Both ships carried the same array of in situ sensors

as the float, minus the ISUS $NO_3$ sensor and the Aandera optode. In addition, the R.V. *Knorr* carried a second CTD and an above-water PAR sensor. Unlike the float, both of the ship's PAR sensors measured scalar PAR. The Seagliders were deployed together with the float and piloted to follow it throughout the experiment. Over the deployment, the distance between the float and individual gliders ranged from 175 km to < 1 km. However, at least one glider was within 50 km of the float for almost the entire deployment, and starting on May 6, all gliders remained within 50 km. Seagliders carried an array

of sensors, but here we only discuss Seaglider estimates of sinking flux, derived in Briggs et al. (2011) using spikes caused by large particles in $b_{bp}$, measured by a WET Labs BB2F.



## 2.2 Discrete sampling

Discrete samples from all three cruises were analyzed at depths ranging from near surface (3-5 m) to 600 m for particulate organic carbon (POC; n=343), chlorophyll *a* (*Chl*; n=935), SiO₄ and NO₃ (n=1001), and phytoplankton pigments (n=80). Detailed methodology for these analyses can be found in the following technical report: http://data.bco-dmo.org/NAB08/Laboratory_analysis_report-NAB08.pdf. Briefly, *Chl* samples were filtered onto GFF 0.7 µm filters and analyzed onboard using a Turner Designs Model 10-AU fluorometer. Following JGOFS protocols, POC samples were filtered onto pre-combusted GFF 0.7 µm filters, sealed in foil packets and stored at -20 °C until analysis onshore using a Perkin Elmer 2400 CHN analyzer. For nutrients, 60 mL samples were immediately frozen and stored at -20°C until analysis onshore using a Latchat Quickchem 8000 Flow Injection Analysis System. In addition, discrete samples on the May process cruise were analyzed for dissolved oxygen concentration via Winkler titrations (n=131) and for bacterial counts and phytoplankton community composition using a FACScan flow cytometer and a FlowCAM automated microscopic imager. Phytoplankton particles were divided into several groups based on optical properties, size, and morphology as described in Cetinić et al. (2012), with more detailed methods in a technical report accompanying the dataset: http://data.bco-dmo.org/NAB08/Phytoplankton_Carbon-NAB08.pdf.

## 2.3 Calibration of in situ sensors

The ship's profiler was held at constant depth for 60 s prior to closing each bottle to capture a water sample. In situ measurements from the 30 s prior to bottle closing were averaged to obtain a single value for matchups with discrete samples. Ship in situ sensors were calibrated via linear regression against discrete measurements. Float in situ sensors were calibrated using data from ten calibration casts, in which the ship was brought to the float's location and both ship and float profiled simultaneously. Float NO₃ and oxygen sensors were calibrated directly against the discrete measurements taken during the calibration casts. All other float sensors were calibrated against the matching ship in situ sensors, in order to maximize the number of matchups. Individual calibration details for each float sensor are listed below.

### 2.3.1 Temperature and Salinity

The duplicate temperature (T) and salinity (S) sensors aboard the ship's profiler during the May process cruise agreed closely (median S difference ≤ 0.0018 and a median T difference ≤0.0006 °C for each of 134 profiles). The ship TS sensors were therefore used as standards, after de-spiking and averaging (more details at http://data.bco-dmo.org/NAB08/Ship_TS_despiking-NAB08.pdf). Duplicate T sensors aboard the float also agreed closely and were therefore combined into a single record without adjusting to match the ship. After reconciliation of duplicate S measurements on each platform, a small mismatch between float and ship salinity was identified from the calibration casts and corrected by subtracting 0.0075 from the float S (more details at http://data.bco-dmo.org/NAB08/CTD_float_Calibration-NAB08.pdf).



### 2.3.2 Oxygen

Comparison between the SBE-43 and optode oxygen sensors aboard the float revealed differing sources of bias in each sensor. Bias in SBE-43 oxygen was introduced by changes in pumping rate during different modes of float operation and by wave action near the surface. Bias in optode oxygen arose from its factory calibration, T and pressure effects, and a slower

time response. After reconciliation of the two sensors to reduce these biases, the SBE-43 oxygen was brought in line with the discrete oxygen samples on the best six calibration casts by subtracting a constant offset of 0.9 µMol kg$^{-1}$. We conclude that the accuracy of the corrected in situ oxygen estimates is better than 2 µMol kg$^{-1}$, based on agreement with discrete samples (Winkler titrations). More details of the float's oxygen calibration can be found at http://data.bco-dmo.org/NAB08/Oxygen_Calibration-NAB08.pdf.

### 2.3.3 POC from optical beam attenuation

Raw output from the float optical beam transmissometer was aligned with raw ship transmissometer output using matchups from eight of the calibration casts. Agreement was very good (r$^2$ = 0.99), showing no evidence of sensor drift. Intercalibrated raw transmissometer output was converted to particulate optical beam attenuation $c_p$ using the mean of factory calibrations performed on the ship's transmissometer before and after deployment. More details can be found at http://data.bco-

dmo.org/NAB08/C-Star_Calibration-NAB08.pdf. We estimated $c_p$-derived POC (POC$_{cp}$) following Cetinic et al. (2012), but with a time-dependent adjustment in POC/$c_p$ ratio to account for community changes. After subtracting the POC/$c_p$ regression offset of 0.015 m$^{-1}$ (Cetinic et al. 2012) from our $c_p$ measurements, we computed the POC/$c_p$ ratio for all ship POC and $c_p$ samples where $c_p$>0.2 m$^{-1}$ in the upper 30 m during the May process cruise (Fig. 2; gray points). Samples whose T, S, $c_p$, and $b_{bp}$ matched the float ML measurements within 0.25°C, salinity of 0.01, 0.1 m$^{-1}$, and 0.001 m$^{-1}$, respectively are

shown as black circles (Fig. 2). Three inflection points were fit by eye at 370, 310 and 450 mg m$^{-2}$ on May 6, 11, and 15, respectively. A continuous estimate of POC/$c_p$ at the float patch was obtained by interpolating between these points and assuming constant POC/$c_p$ before May 6 and after May 15 (Fig. 2, red line). This continuous estimate of POC/$c_p$ was multiplied by float $c_p$ (minus offset of 0.015 m$^{-1}$) to yield a $c_p$-based float POC estimate (POC$_{cp}$).

### 2.3.4 POC from optical backscattering

An average of pre- and post-deployment calibrations were used to convert raw backscattering output from both the float and the ship to the volume scattering function at the angle (140°) and wavelength (700 nm) of the sensors. The volume scattering function of seawater was then calculated following Zhang et al. (2009) and subtracted to yield scattering due to particles. The result was multiplied by 2π$\chi$ to yield the particulate backscattering coefficient $b_{bp}$, where the angle-dependent scale factor $\chi$ is 1.132 for the FLNTU scattering sensors used in this study (M. Twardowski, pers. comm.). Float $b_{bp}$ was aligned

with ship $b_{bp}$ using matchups from eight calibration casts (r$^2$ = 0.96). More details can be found at http://data.bco-dmo.org/NAB08/Backscatter_Calibration-NAB08.pdf. Glider $b_{bp}$ was calibrated against the ship FLNTU in a similar fashion





to the float (Briggs et al. 2011). We estimated $b_{bp}$-derived POC ($POC_{bbp}$) following Cetinic et al. (2012) via the equation $POC_{bbp}$ [mg C m$^{-3}$] = 37500 $b_{bp}$ [m$^{-1}$] – 14, derived from a linear regression between co-located measurements of POC and $b_{bp}$ within the mixed layer from the May process cruise.

### 2.3.5 Chlorophyll *a*

Raw chlorophyll fluorometer output from the ship was converted to an initial *Chl* estimate $Chl_{factory}$ using the factory calibrated scale factor and a dark offset derived from the minimum of all per-cast deep values (defined as the median between 550-580 m). An empirical fit between $Chl_{factory}$, T, PAR, and ship discrete *Chl* measurements was used to derive an in-situ *Chl* product (Eq. 1), which was strongly correlated with discrete *Chl* ($r^2$ = 0.87). Float $Chl_{factory}$ was aligned with ship

$$Chl = Chl_{factory} * \frac{2.1*10^{(T-9.2)*0.8}+0.44}{10^{(T-9.2)*0.8}+1} * \frac{(log_{10}(PAR)*0.05+1.02)*tanh\left(\frac{PAR}{95}*0.55\right)}{0.55*\frac{PAR}{95}}, \tag{1}$$

$Chl_{factory}$ using the matchups from eight calibration casts ($r^2$ = 0.95), allowing calculation of *Chl* via Eq. (1), at the float as well. The FLNTU sensor was located at the bottom of the float, facing down, so *Chl* data was removed whenever the float was moving upward at >1.7 cm s$^{-1}$, due to possible entrainment of deeper water. *Chl* measurements where PAR>75 were also removed to eliminate non-photochemical quenching. In order to obtain a continuous, depth-resolved record of *Chl* for calculation of primary productivity, the remaining *Chl* estimates, from both mixed-layer mode and profiles, were filtered using a 5-point running median, averaged in one-hour, one-meter bins, and then linearly interpolated in depth and time, such that a 30 m vertical interval and a 1 day time interval were considered equidistant.

### 2.3.6 Nitrate

A post-deployment laboratory calibration, including temperature and salinity corrections, was used to obtain initial $NO_3$ estimates from the float's ISUS $NO_3$ sensor. An additional scale factor of 1.15 and offset of +2.6 μM were required to bring these initial estimates in line with discrete samples taken during calibration casts. More details can be found in Alkire et al. (2012) and at http://data.bco-dmo.org/NAB08/ISUS_Nitrate_Calibration-NAB08.pdf.

### 2.3.7 Silicate

$SiO_4$ was not measured by the float, but discrete shipboard $SiO_4$ measurements from the top 15 m were considered to represent mixed-layer $SiO_4$ at the float location if the corresponding temperature, salinity and $NO_3$ measurements matched concurrent float ML measurements to within 0.25°C, salinity of 0.01, and 0.8 mmol m$^{-3}$ respectively.

### 2.3.8 PAR

The factory calibration of the float PAR sensor was used "as is."



### 2.4 Mixed layer depth

Mixed layer depth (MLD) was calculated at hourly intervals from float potential density anomaly estimates via the following steps: 1) Smooth density timeseries using a 5-point running median. 2) Average density into one-hour, one-meter bins. 3) Fill in the gaps with 2D linear interpolation, such that a 30 m vertical interval and a one-day time interval are considered equidistant. 4) For each hour, find the minimum potential density anomaly. 5) The MLD for each hour is defined as the shallowest depth where the potential density anomaly exceeds this minimum by $\geq 0.01$ kg m$^{-3}$. In order to minimize the influence of water entrainment by the float, the MLD was calculated twice, once excluding data when downward velocity exceeded 1 m min$^{-1}$ and once excluding downward velocities exceeding 1 m min$^{-1}$. We use the average of these two estimates as the final MLD estimate. When the float was close to neutral buoyancy, this MLD(t) estimate followed the lower limit of the vertical movement of the float during its ML mode. However, during periods of positive buoyancy, MLD(t) occasionally exceeded the maximum depth of the float during its ML mode (Fig. 3).

### 2.5 K$_{PAR}$

#### 2.5.1 Instantaneous $K_{PAR}$ estimates

The diffuse attenuation coefficient of PAR $K_{PAR}$ was calculated from each pair of consecutive PAR measurements made at times $t_1$ and $t_2$ via Eq. (2), where $z$ is depth, $\bar{z}$ is the mean of z(t$_2$) and z(t$_1$), and $\bar{t}$ is the mean of t$_2$ and t$_1$.

$$K_{PAR(measured)}(\bar{z}, \bar{t}) = \frac{ln(PAR(t_1)) - ln(PAR(t_2))}{z(t_2) - z(t_1)} \tag{2}$$

#### 2.5.2 K$_{PAR}$ fit method

The uncertainty of individual $K_{PAR(measured)}$ estimates was high and depended strongly on d$z$, which ranged from 0.2 – 30 m with a mean of 1.3 m. These 14000 $K_{PAR(measured)}$ estimates were therefore fit to *Chl* and $z$ using a non-linear least-squares multiple regression weighted by d$z$ to obtain Eq. (3):

$$K_{PAR(modeled)}(Chl, z) = 0.064 * Chl^{0.51} + 0.20 * max(z, 2.5)^{-0.63} + 0.0031. \tag{3}$$

In order to evaluate the performance of this fit, in-situ $K_{PAR}$ precision was increased by eliminating estimates with d$z$<2 m and combining the remaining estimates into 21-point medians, yielding a total of 118 independent in-situ $K_{PAR}$ estimates. A type-II linear regression of these estimates against 21-point medians of $K_{PAR}$ estimated via Eq. (3|) yielded an r$^2$ of 0.85, a root mean square error of 0.014 m$^{-1}$, and a mean bias of -0.004 m$^{-1}$. The residual error was not significantly correlated with depth, time, solar zenith angle or the ratio of in situ *Chl* to $b_{bp}$, a proxy for plankton community in this system (Cetinić et al. 2015).



## 2.6 Depth-resolved PAR

In order to calculate PAR at all depths, PAR was extrapolated from a reference depth $z_{ref}$ via Eq. (4):

$$PAR_{extrapolated}(z) = PAR(z_{ref}) * exp\left(\int_z^{zref} K_{PAR}\, dz\right),\qquad(4)$$

using $K_{PAR(modelled)}$ calculated via Eq. (3) from the float's continuous *Chl*. When the float was within the top 50 m, $z_{ref}$ was the

5 depth of the float and $PAR(z_{ref})$ was the float's PAR measurement. The performance of this extrapolation was evaluated by comparing $PAR_{extrapolated}(0-)$ (just below the surface) calculated via Eq. (4) with scalar $PAR(0+)$ measured by the ship's underway system. For all measurements where the ship was within 1 km of the float, the float was in the top 50 m, and $PAR(0+)$ was greater than 1 µmol m$^{-2}$ s$^{-1}$, $PAR(0+)$ and $PAR_{extrapolated}(0-)$ were highly correlated ($r^2 = 0.96$ on a linear scale and $r^2 = 0.99$ on a logarithmic scale). The geometric mean of the ratio of $PAR_{extrapolated}(0-)$ to $PAR(0+)$ was 0.92 and the

10 geometric (multiplicative) standard deviation was a factor of 1.19. For several hours each afternoon, while the float profiled to 250 m, float PAR measurements were not available, so $PAR(0-)$ was estimated using an empirical function of solar zenith angle and an empirical index of cloud cover. First, a double exponential was fit to 36000 PAR measurements obtained in the top 1 m over a range of solar zenith angles from -6° to 90° by a global network of 100 "Biogeochemical Argo" type profiling floats to obtain $PAR_{modelled}(0-)$, an estimate of $PAR(0-)$ under mean cloud and atmospheric conditions:

$$log_{10}(PAR_{modeled}(\theta)) = 2.5 * exp(0.0030 * \theta) - 1.7 * exp(-0.10 * \theta),\qquad(5)$$

To adjust for clouds, $PAR_{extrapolated}(0-)$ from the Lagrangian float (via Eq. 4) was divided by corresponding estimates $PAR_{modelled}(0-)$ from to obtain an index of sunniness, which was averaged into 15 min bins to remove noise from wave focusing. This sunniness index ranged from 0.1 to 3.6 over the entire float deployment. Sunniness index at time $t$ was estimated using a ±1-day running mean of these sunniness index estimates, weighted by the inverse square of $t$-$t_i$, where $t_i$ is

20 the time of each measurement. This running mean sunniness index was then multiplied by $PAR_{modelled}(0-)$ to obtain $PAR_{adjusted}(0-)$, which was used as $PAR(z_{ref})$ in Eq. (4) during the afternoon gaps.

## 2.7 O$_2$ airsea flux

O$_2$ airsea flux was calculated following Alkire et al. (2012). Briefly, wind speeds were taken from the NCEP WW3 Global Reanalysis product, except during the May cruise, when ship wind measurements were used. O$_2$ saturation was calculated

following García and Gordon (1992). Airsea flux was calculated following (Wanninkhof 1992), modified to account for bubble injection following Woolf and Thorpe (1991). Hourly dO$_2$/dt in the ML due to airsea flux was estimated by dividing hourly flux estimates by hourly MLD.

## 2.8 Primary Productivity estimates

### 2.8.1 Diel cycles of O$_2$ and POC

"Typical" diel cycles (minimum near dawn and maximum near dusk) were observed in mixed layer records of O$_2$ (Fig. 4), consistent with previous studies (Caffrey 2003; Hamme et al. 2012; Nicholson et al. 2015). We estimated mixed layer gross





oxygen productivity (GOP) at half-day intervals from these diel cycles. To estimate morning GOP, ML $O_2$ concentrations were smoothed with a 3-point running median and a type I linear regression ($O_2$ vs time) was fit to data from dusk to dawn (Fig. 4a; solid black line). The regression fit was projected forward to provide an estimate of noontime mixed-layer $O_2$ in the absence of GOP. Measured noontime $O_2$ was calculated from a type I linear regression of $O_2$ data taken within 1 h of local

noon. Morning mixed layer GOP was calculated as the difference between measured and projected concentration (Fig. 4; blue vertical bar) and divided by 0.5 d to convert to units of mmol $m^{-3}$ $d^{-1}$. Afternoon GOP was calculated in a similar fashion, by subtracting noontime mixed-layer $O_2$ from the noontime extrapolation of a linear fit of the following night's data. Similar diel cycles were observed in mixed layer $POC_{cp}$, and the same method was used to calculate mixed layer gross primary productivity of POC ($GPP_{cp}$) from these cycles (Fig. 4b). Diel cycles in $POC_{bbp}$ were less regular and usually out of

phase with $O_2$ and $POC_{cp}$ cycles, but $GPP_{bbp}$ was calculated in the same way as GOP and $GPP_{cp}$ for comparison. Note that this diel cycles method assumes homogeneous mixing to a constant depth and that any gain or loss terms other than GOP (or GPP) are constant day-to-night over the period of a single calculation (~18 h). However, we find a clear diel cycle in MLD (Fig. 3), which amplifies the diel cycle in $O_2$ (and $c_p$ and $b_{bp}$), causing PP calculated from diel cycles to exceed mean PP within the daily mean MLD. Analysis of the output of a coupled physical-biological model assimilating data from the

Lagrangian float (Bagniewski et al. 2011), which accurately reproduced the diel cycle in mixing (Fig. 3, black line) shows that the mixing-amplified diel cycles of $O_2$ in the ML yield daily GOP estimates that correspond approximately to the mean GOP above the daily *minimum* MLD. Regression of diel GOP as a function of "true" model GOP, forced through zero, yields a slope ±95 % confidence interval of 0.91±0.12 and a RMSE of 0.12 mmol $m^{-3}$ $d^{-1}$. Bias in GOP due to day-night differences in airsea flux was also estimated using the difference between mean morning (or afternoon) $dO_2/dt$ due to airsea

flux and that of the previous (or next) nighttime. Mean bias was small (<5% of GOP), and linked primarily to the MLD diel cycle, so a separate correction was not deemed necessary. Other potential biases are discussed in sections 4.1.2 to 4.1.4.

### 2.8.2 [14]C incubations

During the April and May cruises, daily two-hour [14]C incubation experiments were conducted (n=28) to estimate photosynthetic parameters. Each day, a water sample was taken from the *Chl* maximum, as determined by in situ

fluorescence, and duplicate 2 h [14]C incubations were performed at 7 different PAR levels ranging from 0-400 µmol $m^{-2}$ $s^{-1}$. The dark incubation [14]C activities were weakly but significantly correlated with *Chl* (type II linear regression; $r^2 = 0.19$; p<0.05; apparent NPP = *Chl* * 0.036±0.026 mg C mg $Chl^{-1}$ $h^{-1}$ + 0.049±0.035 mg C $m^{-3}$ $h^{-1}$); dark activities were treated as sample-specific blanks and subtracted from the light incubation activities of the corresponding water sample. The resulting productivity estimates were interpreted as net primary productivity (NPP), based on findings that most phytoplankton do not

respire old carbon when newly fixed carbon is available (Marra and Barber 2004; Pei and Laws 2013). Note that if, contrary to our assumptions, phytoplankton did respire old carbon at all light levels during these incubations, then our calculations below overestimate NPP and underestimate phytoplankton respiration ($R_\Phi$), but GPP is unbiased. On the other hand, if old




carbon is respired only in the low light incubations, then we under-estimate $R_\Phi$ and GPP, but little bias is introduced in NPP. Seven of the 350 individual NPP estimates (all from the April cruise) were judged to be positive outliers and were manually removed before further analysis. In ~60% of the incubation experiments, NPP decreased with increasing PAR for PAR>200 µmol quanta m$^{-2}$s$^{-1}$. We conclude that this apparent photoinhibition is likely not representative of most field conditions,

because in situ measurements of *Chl*-normalized dO$_2$/dt showed no consistent relationship with PAR between PAR values of 100 and 1000 µmol m$^{-2}$s$^{-1}$. We therefore removed values of NPP where PAR > 200 µmol m$^{-2}$s$^{-1}$ if they were lower than the second-highest NPP observed where PAR < 200 µmol m$^{-2}$s$^{-1}$ (56 of 110 high light points removed). Remaining NPP vs PAR data were fit to an empirical "PvE" model represented in Eqs. (6-8):

$$\lambda = PAR \frac{\alpha}{P_M} \varepsilon, \tag{6}$$

$$GPP = P_M \left( 1 + \frac{1}{\varepsilon} \sum_{i=0}^{\varepsilon-1} e^{-\lambda} \frac{\lambda^i}{i!} i - \sum_{i=0}^{\varepsilon-1} e^{-\lambda} \frac{\lambda^i}{i!} \right), \tag{7}$$

$$NPP = GPP - R_\phi, \tag{8}$$

based on four parameters: maximum GPP ($P_m$), the initial slope of GPP/PAR ($\alpha$), $R_\Phi$, and an efficiency factor ($\varepsilon$) representing "sharpness" of the transition between light-limited and light-saturated photosynthesis. We used a single $\varepsilon$ value of six, which provided the best overall least squared fit across all incubation experiments. This ε value yields a NPP vs PAR

relationship that is "sharper" than the commonly used "tanh" model (Harrison and Platt 1986) and more linear at low PAR, leading to smaller y-offset (smaller $R_\Phi$ estimate). See Fig. 5 for example fits. A power law was then fit between in situ *Chl* estimates from the ship's profiling package (calculated via Eq. 1) and each of the three parameters obtained from each NPP vs PAR fit ($P_M$: Fig. 6a; $\alpha$: Fig. 6b; and $R_\Phi$: Fig. 6c). Fits with $P_M$ and $\alpha$ used data from all cruises, but the fit with $R_\Phi$ included only data from the process cruise (Fig. 6c; solid circles), as signals were too low to constrain $R_\Phi$ in April and $R_\Phi$

appeared consistently higher during the June cruise, possibly due to higher temperature.

**2.8.3 Chl-based GPP and NPP**

The relationships in Fig. 6 were used to estimate photosynthetic parameters $P_M$(t,z), $\alpha$(t,z), and $R_\Phi$(t,z) and their uncertainty intervals at the float location from *Chl*(t,z) (Section 2.3.5). We estimated gross primary productivity GPP$_{Chl}$(t,z) and net primary productivity NPP$_{Chl}$(t,z) via Eqs. (6-8) using the above photosynthetic parameters, PAR$_{extrapolated}$(t,z) (Section 2.6),

and ε=6 as input. Uncertainties were propagated from $P_M$(t,z) $\alpha$(t,z), and $R_\Phi$(t,z) using the conservative assumption that they covary (i.e. upper bound of NPP$_{Chl}$ was derived from upper bounds of $P_M$ and $\alpha$ and lower bound of $R_\Phi$).

**2.9 Area-weighted mean particle diameter**

Area-weighted mean particle diameter $D_{bbp}$ 10-50 m depth bin was estimated following Briggs et al. (2013) via Eqs. (9-11):

$$D_{bbp} = 2 \sqrt{\frac{Var[b_{bp}(t)]}{E[b_{bp}(t)]} \frac{V}{Q_{bb}} \frac{1}{\gamma(\tau)} \frac{1}{\pi}}, \tag{9}$$



$$\gamma(\tau) = \begin{cases} 1 - (3\tau)^{-1}, & if\ \tau \geq 1 \\ \tau - \tau^2/3, & if\ \tau \leq 1 \end{cases}, \tag{10}$$

$$\tau = \left( \frac{t_{res}}{t_{samp}} \right), \tag{11}$$

where $Var[b_{bp}(t)]$ is the variance in $b_{bp}$ due to random distribution of particles in space, $E[b_{bp}(t)]$ is mean $b_{bp}$, $V$ is sensor sample volume, $Q_{bb}$ is the backscattering efficiency, and $\gamma$ and $\tau$ are functions of residence time in the sample volume $t_{res}$ and

sample integration time $t_{samp}$. $Var[b_{bp}(t)]$ and $E[b_{bp}(t)]$ were calculated once per profile (ascent or descent) using all data between 10-50 m. Prior to calculation of $Var[b_{bp}(t)]$, the $b_{bp}$ timeseries was de-trended by subtracting a 7-point running median and large outliers (greater than five times the interquartile range) were removed before the variance was calculated on the residuals. A $V$ of 0.62 ml was used (Briggs et al. 2013) and a $Q_{bb}$ of 0.02 was assumed (based on empirical $b_{bp}/c_p$ ratio of ~0.01 and theoretical value of $Q_c$=2 for diameter>>wavelength; Bohren and Huffman, 1983). A $t_{res}$ of 0.02 s was chosen

based on a 6 mm path through the sample volume and a platform velocity of 30 cm s$^{-1}$ and $t_{samp}$ was 1 s.

## 2.10 Sinking POC flux

POC$_{bbp}$ profiles from both gliders and the float were divided into a "small" particle baseline (7-point running minimum followed by running maximum) and a "large" particle "spike" signal (residuals above the baseline). Large particle POC$_{bbp}$ was multiplied by a bulk sinking speed of 75 m d$^{-1}$ to estimate large POC flux (Briggs et al. 2011). A broad plausible range

of bulk sinking speeds 5 ± 5 m d$^{-1}$ was used to estimate small POC sinking flux, which was added to large POC flux to yield total sinking POC flux. Sinking POC flux was bin averaged in 50 m vertical bins and either running 2-day bins (gliders) or longer discrete bins to match bloom stages (float).

## 3 Results

### 3.1 Evolution of the spring bloom

From float deployment through April 17, MLD was variable (often >200 m but occasionally below 50 m; Fig. 3), mixed-layer nutrients were high (NO$_3$ ≈ 12 mmol m$^{-3}$; SiO$_4$ ≈ 4 mmol m$^{-3}$), biomass was low (*Chl* ≈ 0.35 mg m$^{-3}$; POC$_{cp}$ ≈ 35 mg m$^{-3}$), and O$_2$ was undersaturated by ~ 10 mmol m$^{-3}$ (Fig. 7). Mixed-layer biomass concentrations increased over the next month, peaking in mid-May. This broad increase was punctuated by several 1-2 day periods of decrease, most associated with clear mixed-layer deepening (Fig. 7). SiO$_4$ was depleted to its lowest level on May 11, *Chl* concentration peaked on

May 12, and NO$_3$ depletion and POC$_{cp}$ and O$_2$ concentrations peaked on May 13. From bloom peak to May 16, *Chl* decreased dramatically (77%), POC$_{cp}$ and O$_2$ decreased moderately (by 9 and 13 mmol m$^{-3}$, respectively), and NO$_3$ and SiO$_4$ concentrations recovered slightly (by 0.8 and 0.4 mmol m$^{-3}$, respectively).



## 3.2 Primary productivity estimates

All GPP and GOP estimates were averaged into 3-day bins to improve precision of the diel-cycles-based estimates. To first order, $GPP_{Chl}$ followed *Chl*, low in early April (0.5-1.0 mmol m$^{-3}$ d$^{-1}$), peaking near 10 mmol m$^{-3}$ d$^{-1}$ between May 7 and 13, then decreasing to near 3 mmol m$^{-3}$ d$^{-1}$ or below after the bloom (Fig. 8). But increases in $GPP_{Chl}$ led increases in Chl by 1-2

5   days during ML shoaling (and high growth) events on April 24-27 and May 6-8. For the entire "bloom growth" phase from early April through May 9, $GPP_{Chl}$ was strongly correlated with both cycle-based estimates of both GOP (Fig. 8b and Fig. 9b; blue) and $GPP_{cp}$ (Fig. 8c and Fig. 9c; blue). GOP was a factor of 2.1 higher than $GPP_{Chl}$ on a molar basis, while $GPP_{cp}$ was slightly lower (factor of 0.81). $GPP_{bbp}$ was poorly correlated with $GPP_{Chl}$ (Fig. 8d and Fig. 9d; blue) and significantly lower (by 60%; Fig. 9d). From noon May 10 to noon May 11, diel cycles could not be calculated, because the float was

trapped at the surface, due to high stratification and slight positive buoyancy. At peak biomass (May 11-13), and the bloom decline (May 13-16), both $GOP/GPP_{Chl}$ and $GPP_{cp}/GPP_{Chl}$ were substantially lower (Fig. 8, pink highlighted region, and Fig. 9; pink symbols). In the post-bloom period (May 16-24), $GOP/GPP_{Chl}$ and $GPP_{cp}/GPP_{Chl}$ increased again, similar to the bloom growth ratios (Fig. 8 and Fig. 9; red symbols). When all bloom phases are combined, best-fit ratios of GOP and $GPP_{cp}$ to $GPP_{Chl}$ are 1.7 and 0.6, respectively and correlations are considerably less strong ($r^2$ of 0.67 and 0.49, respectively).

However, the estimates of productivity from diel cycles (GOP and $GPP_{cp}$) remained strongly correlated for the entire deployment. Over the entire study period, morning estimates of GOP and $GPP_{cp}$ were not significantly different from the afternoon estimates, while morning $GPP_{bbp}$ estimates were significantly lower than afternoon estimates (80% lower overall). However, morning-afternoon patterns appear to change starting on May 13, when the bloom decline starts (e.g. Fig. 4). From May 13-24, there is no significant difference between morning and afternoon $GPP_{bbp}$, but afternoon estimates of $GPP_{cp}$ and

GOP, were lower than morning estimates by 70% and 43%, respectively. These differences were near the threshold of statistical significance: mean afternoon-morning difference ± 2 standard errors was -2.3±2.3 mmol m$^{-3}$ d$^{-1}$ for $GPP_{cp}$ and -3.0±2.5 mmol m$^{-3}$ d$^{-1}$ for GOP.

## 3.3 Depth integrated GPP, NPP, and NCP and carbon export

Alkire at al. (2012) estimated depth-integrated net community productivity (NCP) integrated within the top 50-60 m and

carbon export from 50-60 m at the float location for four periods of stable stratification: the "early bloom" (April 23-27), "main bloom" (May 6-13), "decline" (May 13-14) and "post bloom" (May 20-24). We integrated $GPP_{Chl}$ and $NPP_{Chl}$ to the same depth and time ranges in order to assemble detailed organic carbon budgets for these periods (Fig. 10). Each budget term carries considerable uncertainty, but based on the central estimates, the partitioning of fixed carbon appeared to change substantially over the course of the bloom. Note that these NCP estimates include net production of dissolved organic carbon

(DOC), while $NPP_{Chl}$ excludes any photosynthetic DOC production. $NPP_{Chl}$ and NCP estimates were similar during the early and main bloom, suggesting moderate-to low heterotrophic respiration. During the early bloom period, export was also low (~22-28% of $GPP_{Chl}$), allowing rapid accumulation of biomass. During the main bloom, $GPP_{Chl}$ nearly doubled as biomass





increased, but a larger fraction (~50%) was exported, leaving ~25% to accumulate. During the bloom decline, apparent community respiration was 156% of GPP$_{Chl}$ and export was an additional 50-80%. In the post-bloom period, community respiration was again high (~100% of GPP), and export was much lower (0-15% of GPP). Our NPP$_{Chl}$ estimates and $b_{bp}$ "spike"-based sinking flux estimates provide a continuous, high-resolution picture of the link between productivity and export at 125 m for the entire study period (Fig. 11a). Float and glider-based POC export estimates agree broadly at this depth (red lines), suggesting that the higher-resolution glider timeseries are representative of the float patch as well. While export at 125 m is coupled with NPP$_{Chl}$ (Fig. 11a), there is a rapid increase in export efficiency between May 3-6 from ~20% to 40%. Area-weighted mean particle diameter ($D_{bbp}$) ranged from 90-150 µm during April, peaked at 250 µm on May 7-8 (Fig. 11b), coincident with peak biomass as measured by both $Chl$ and POC$_{bbp}$ from the gliders (not shown). $D_{bbp}$ fell rapidly on May 9, coincident with a ML deepening event. Post-bloom $D_{bbp}$ ranged from 150-190 µm (Fig. 11b).

## 4 Discussion

### 4.1 Accuracy of PP estimates

The combination of three estimates of primary productivity and one estimate of community productivity, all from the same platform at comparable temporal and horizontal scales, provides a unique opportunity to evaluate the accuracy of all methods. Each of our PP methods is discussed in turn in sections 4.1.1-4.1.4.

### 4.1.1 GPP$_{Chl}$

GPP$_{Chl}$ and GPP$_{cp}$ are estimates of the same quantity, obtained independently. GPP$_{Chl}$ is derived from PAR and $Chl$ estimates using robust local parameterizations obtained from $^{14}$C incubations. GPP$_{cp}$ is derived entirely from $c_p$ measurements, converted to POC using another robust, local empirical relationship. The averaging depth (daily minimum MLD) for GPP$_{Chl}$ was chosen to match the diel cycles method based on results of a model tuned to match local conditions (Bagniewski et al. 2011). In this context, the combination of strong correlation and absolute agreement between GPP$_{Chl}$ and GPP$_{cp}$ (within 18%) provides confidence in both methods during the bloom growth and post bloom periods. The GOP/GPP$_{Chl}$ slope of 2.2 is at the upper end of the expected range, providing additional first-order support for GPP$_{Chl}$ accuracy. Neither GPP method includes DOC production, so the range of expected photosynthetic quotients (~1-1.45; Laws 1991; Robertson et al. 1993), combined with the fraction of GPP released as DOC in marine/estuarine environments (2-50%; Baines and Pace, 1991) imply a possible GOP/GPP range of 1-2.9. During the main bloom observed by the float in this study, Alkire et al. (2012) estimate that DOC accounts for 22-40% of NCP in the mixed layer during the main bloom. If these estimates apply to GPP as well, our expected GOP/GPP range narrows to 1.3-2.4. Thus, our GOP estimates suggest either that both the photosynthetic quotient and phytoplankton DOC production are high during bloom growth (and GPP is accurate) or that both GPP estimates are biased low. As mentioned in section 2.8.2, a negative bias in GPP$_{model}$ could be explained if phytoplankton



respired substantial old, unlabelled carbon in our low light incubations, but not in the high light incubations. In this case a separate explanation is needed for the high GOP/GPP$_{cp}$ slope of 2.7 (see next section).

During bloom peak and decline, the strong discrepancies between GPP$_{cp}$ and GPP$_{Chl}$ imply either an underestimate by GPP$_{cp}$

(discussed in the next section) or an over-estimate by GPP$_{Chl}$. If diatoms reduce GPP in response to sustained SiO$_4$ limitation then we expect GPP$_{Chl}$ over-estimation at peak biomass, given that GPP$_{Chl}$ is only a function of *Chl* and PAR, without any nutrient limitation term. Twelve mixed-layer SiO$_4$ samples were collected in the vicinity of the float on May 11-13, and the mean and maximum measured concentrations were 0.3, and 0.6 mmol m$^{-3}$, respectively, suggesting that diatom growth was most likely severely limited (Fig. 7a). This does not necessarily imply that diatom carbon fixation rates were reduced, but

previous studies have indeed observed a large and reversible reduction in apparent diatom photosynthetic efficiency under multi-day SiO$_4$ limitation (Lippemeier, Hartig, and Colijn 1999; Lippemeier et al. 2001). Both FlowCAM microscopy and HPLC pigments indicate that diatoms accounted for $\geq$ 50% of phytoplankton biomass at bloom peak (Cetinić et al. 2015), so we expect a substantial reduction in bulk phytoplankton growth (and likely GPP) under these conditions. This expectation, combined with the observed reduction in both GPP$_{cp}$ and GOP at bloom peak, leads us to conclude that GPP$_{Chl}$ is most likely

over-estimated at bloom peak. This conclusion agrees with the coupled physical-biological model of Bagniewski et al. (2011), which assimilated float biogeochemical measurements achieved optimal fit when diatom GPP was limited by SiO$_4$ with a half-saturation constant of 1 μmol m$^{-3}$. GPP inferred from this model closely matches our observed GPP$_{cp}$ during SiO$_4$ limitation (Fig. 8, gray line vs black circles), even though Bagniewski et al. (2011) assimilated daily binned data, removing any diel cycle information. On the other hand, three $^{14}$C incubations were conducted between May 10-14 using water with

SiO$_4$<0.5 mmol m$^{-3}$, although not at the float location, and there was not a substantial reduction in measured P$_M$. These samples may not be representative of the water sampled by the float, despite similar *Chl* and SiO$_4$ concentrations, or it is possible that a bottle effect enhanced GPP. But we cannot rule out the alternative hypothesis that the Si-limited community continued to fix carbon at a constant rate, and that GPP$_{cp}$ and GOP estimates were reduced for another reason (discussed in next sections).

Apart from Si limitation, possible explanations for GPP$_{Chl}$ over-estimation include over-estimation of *Chl* due to increased fluorescence, underestimation of the MLD, or photoinhibition. Again, *Chl* was calculated the same way for the floats and ship, so if the in-situ fluorometric method over-estimated *Chl* at bloom peak, we would expect to see a deviation from the observed relationships between photosynthetic parameters and *Chl* (Fig. 6). So this explanation, while plausible given the

30 high *Chl*/b$_{bp}$ ratio at bloom peak (Cetinić et al. 2015), also requires that none of our low SiO$_4$ bottle samples were representative of the bloom peak at the float location. The density-based MLD estimates appear quite robust during this period, consistently shallow and stable at 10-20 m and matched by the vertical motion of the float. And the daytime increases in O$_2$ (Fig. 4) and $c_p$ on May 11 and 12 show no sign of photoinhibition, despite a peak hourly-averaged PAR of >750 μmol m$^{-2}$s$^{-1}$; increases are smooth throughout the morning and appear to continue at the same rate in the afternoon (Fig. 4).



$SiO_4$ limitation may also explain the some of the discrepancy between $GPP_{cp}$ and $GPP_{Chl}$ during the bloom decline (May 13-16), at least when afternoon estimates are excluded (see Fig. 9b,c; open pink circles). Lower afternoon $GPP_{cp}$ and GOP, combined with very shallow (<5 m) MLDs at noon on May 13 and 15, also raise the possibility of significant photodamage

inhibiting afternoon productivity. Mean ML PAR exceeded 500 µmol m$^{-2}$s$^{-1}$ for at least two hours on both days. However, the negative afternoon $GPP_{cp}$ estimates at this time suggest a bias in the diel cycles method as well (see next section).

### 4.1.2 $GPP_{cp}$

Potential sources of bias unique to $GPP_{cp}$ include a diel cycle in grazing (e.g. due to diel migration of zooplankton), a diel cycle in export loss, or a diel cycle in the $POC/c_p$ ratio. The tight correlations between $GPP_{cp}$ and GOP throughout the entire

study period ($r^2$=0.95; Fig. 9a) and between $GPP_{cp}$ and $GPP_{Chl}$ during bloom growth ($r^2$=0.96; Fig. 9b) provide encouraging support for $GPP_{cp}$ as a measure of *relative* primary productivity at the very least. Furthermore, the quantitative agreement between $GPP_{cp}$ and $GPP_{Chl}$ during both bloom growth and post-bloom (Fig. 9c; slope: 0.82±0.06) is very close to our expected slope of 0.93 from model results (reanalysis of Bagniewski et al. 2011), suggesting that $GPP_{cp}$ accuracy is comparable to other methods across most of the conditions encountered. These findings provide important support for the

method, because, to our knowledge, this is the first time that GPP has been derived from beam transmissometer data and independently validated with the same quantity on the same spatiotemporal scale and across a wide dynamic range. However, it should be noted that our results do not necessarily apply to other systems, where different phytoplankton size and/or timing of cell division could alter the diel $POC/c_p$ relationship (Dall'Olmo, et al. 2011).

If the $SiO_4$ limitation hypothesis is correct, then $GPP_{cp}$ during the bloom peak and morning $GPP_{cp}$ during the bloom decline may be accurate as well. On the other hand, if $GPP_{Chl}$ is accurate during this time, then $GPP_{cp}$ is biased low by ~50% at this time. It is unclear what might cause such a low bias, especially at bloom peak. Between the afternoon of May 11 and the morning of May 13, there is no anomaly in the diel cycles of $POC_{cp}$ or $O_2$ (Fig. 10) indicative of daytime mixing, advection, or possible photoinhibition, and there is no change in the relationship between $GPP_{cp}$ and GOP (Fig. 9a; rightmost pink

symbol). Without grazing data, we cannot rule out enhanced daytime grazing as a possible explanation, although grazing is generally expected to be higher at night. Alternatively, particularly high photo-oxidation could potentially dampen $O_2$ diel cycles during this period and perhaps $c_p$ diel cycles as well. This hypothesis is supported by laboratory measurements of diatom productivity under nutrient limitation (Spilling et al. 2015), although again we would need to explain why reduced $P_M$ was not observed in our bottle incubations. On the other hand, the afternoon $GPP_{cp}$ estimates during the bloom decline

period show a clear example of negative bias in the diel cycles method. On May 13, 14, and 15 (bloom decline), rates of net $POC_{cp}$ (and $O_2$) accumulation are positive or near zero in the morning but negative each afternoon (e.g. Fig. 4; May 13). One plausible explanation is advection of the float relative to the ML during its afternoon profile. During this period, comparison with ship, autonomous glider, and satellite measurements (Alkire et al. 2012) shows that the float was at the edge of a high



biomass (and $O_2$) patch, so advection during this time would most likely cause loss in $POC_{cp}$ and $O_2$. Note that the afternoon reductions in $GPP_{cp}$ during bloom decline are greater than the afternoon reductions in GOP (Fig. 8b,c). This result is possible with the horizontal advection/mixing hypothesis alone, but high export, combined with a shallow afternoon MLD may also play a role. Shallow MLD enhances the loss of ML concentration for a given export rate, and night-time mixing can re-

entrain some of this export, reducing the ML POC diel cycle relative to the $O_2$ diel cycle.

### 4.1.3 GOP

The tight fit between GOP and both GPP estimates over most of the study period provides important support for the $O_2$ diel cycles method as a measure of relative primary production in this region. Again, because all estimates were independent and taken at the same scale, and because the two-month deployment allowed 14 independent matchups at 3-day timescale,

spanning a wide range of productivities, this dataset represents the most extensive validation to date of the $O_2$ diel cycles method as a measure of *relative* primary productivity. Additionally, the overall *accuracy* of our GOP estimates may be assessed indirectly through comparison with independent ship-based GOP estimates made during the May process cruise (Quay et al., 2012) and through comparison of our GOP/GPP and GOP/NPP ratio estimates with previous estimates from this region. Quay et al. (2012) estimated ML-integrated GOP using measurements of three oxygen isotopes: $^{16}O$, $^{17}O$, and

$^{18}O$, taken daily between May 3 and 21 during the process cruise. Mean GOP calculated by this method was 245 mmol $O_2$ m$^{-2}$ d$^{-1}$. This method integrates over several weeks, so we interpret their estimate to correspond roughly to mean ML depth-integrated GOP between April 19 (2 weeks before the first sample) and May 21. For comparison, we multiply each half-day GOP estimate by MLD to obtain ML-integrated GOP and obtain an average from April 19 to May 21 of 149 mmol m$^{-2}$ d$^{-1}$, 40% lower than Quay et al. (2012)'s estimate. However, our estimate integrates to the daily minimum MLD, and while the

triple $O_2$ isotope method assumes constant MLD, we expect it to more closely approximate daily maximum MLD in the presence of diel MLD fluctuations, given its long integration time. Mean $GPP_{Chl}$ during this period, integrated to the bottom of the daily *minimum* MLD is 30% lower than mean $GPP_{Chl}$ integrated to the daily *maximum* MLD. If we assume the same relative difference for GOP, we obtain a revised ML-integrated GOP estimate of 213 mmol m$^{-2}$ d$^{-1}$, 13% lower than Quay et al. (2012)'s estimate. Given the uncertainties associated with the GOP methods as well as the differing spatio-temporal

scales, this result provides first-order support for the accuracy of both methods. Our findings reinforce those of Hamme et al. (2012), who, in the Southern Ocean in March/April, found that mean ML integrated GOP calculated via $O_2/Ar$ diel cycles (similar to our method) was 18% lower than GOP calculated via triple oxygen isotope method (similar to Quay et al., 2012).

Bender et al. (1992) calculated a GOP/NPP ratio of 2.5 during the spring bloom in the Northeast Atlantic, using in situ $^{18}O$

incubations and 24h $^{14}C$ incubations. We calculate $GOP/NPP_{Chl}$ as shown in Fig. 9b, but replace $GPP_{Chl}$ with $NPP_{Chl}$ and obtain a best-fit ratio and 95% confidence interval of 2.4±0.2 for the bloom growth period and 1.7±0.4 for the entire deployment. These fits appear to support the accuracy of both our $NPP_{Chl}$ and GOP estimates during the bloom growth phase, consistent with our other findings. However, our GOP/GPP ratio estimates of 2.6±0.2 (Fig. 9a) and 2.1±0.2 (Fig. 9b)



are near or above the high end of our expected range of 1.3-2.4 (see section 4.1.1). As discussed in previous sections, these ratios may be the result of high photosynthetic quotient and high DOC production, combined with a small negative bias in GPP$_{cp}$. Our GOP estimates may also be too high, but we cannot think of a plausible mechanism that would cause a substantial over-estimate of diel-based GOP (but not of GPP$_{cp}$). Regardless of the source of our high GOP/GPP ratios, they

are also consistent with Hamme et al. (2012), who also estimated GPP from on-deck PvE incubations as well as GOP via O$_2$/Ar diel cycles, providing a very close methodological comparison in a different environment (autumn, Southern Ocean). They obtain an even higher GOP/GPP ratio of 3.6. However, Hamme et al. (2012) assumed that 1-2h $^{14}$C incubations represent GPP, while we assume that these same incubations represent NPP (when NPP>0). If our assumption is correct, then their method provides a quantity closer to daytime NPP than GPP. However, even in this case, assuming moderate

daytime phytoplankton respiration rates (≤30% of GPP), GOP/GPP during their study was >2.5, in agreement with our estimates.

In total, the available evidence provides first-order support for the accuracy of our diel cycles-based GOP estimates. Our findings build on important recent work in diverse environments showing that diel cycles in O$_2$/Ar ratio yield ML GOP

estimates that are consistent with independent GOP estimates (Hamme et al. 2012), and that diel cycles in O$_2$ measurements from autonomous gliders in the subtropical Pacific provide GOP estimates that are a reasonable multiple of independent NPP results (Nicholson et al. 2015). Our results add a third ocean region (springtime North Atlantic) and a third platform (Lagrangian mixed-layer float), in addition to new comparisons with $c_p$ and $b_{bp}$ diel cycles.

### 4.1.4 GPP$_{bbp}$

Because diurnal variability in $b_{bp}$ can be estimated from geostationary satellites (Neukermans et al. 2012), the ability to accurately estimate GPP from $b_{bp}$ diel cycles would be extremely valuable. While ship-based measurements from NAB08 show that $b_{bp}$ and $c_p$ were equally well correlated with POC over the May cruise (Cetinić et al. 2012), the poor matchups we find between GPP$_{Chl}$ and GPP$_{bbp}$, particularly the morning estimates (Fig. 8d), suggest that diel changes are present in POC/$b_{bp}$ and can cause strong, consistent bias in GPP$_{bbp}$. Our results agree with previous findings that while beam

attenuation and forward scattering by phytoplankton increase immediately after they begin to photosynthesize, $b_{bp}$ and side scattering often do not, both in the lab (Ackleson et al. 1993) and in the ocean (Kheireddine and Antoine 2014). These results caution against the use of $b_{bp}$ diel cycles to estimate GPP without further research. However, it is worth noting that our afternoon GPP$_{bbp}$ estimates are reasonably well correlated with GPP$_{Chl}$ ($r^2$=0.63, m = 0.75±0.23; data not shown) during the bloom growth period. If this result is found to be robust in other times and places, then a useful estimate of GPP from

satellite (and other) $b_{bp}$ timeseries may be possible. However, even if the $b_{bp}$ diel cycle cannot be used to estimate GPP, it likely contains other useful information, especially in combination with $c_p$ and/or O$_2$. If robust relationships between plankton community and/or physiology and $b_{bp}$ diel cycles can be established (and, ideally, understood mechanistically),

then measurements of $b_{bp}$ diel cycles may still provide valuable oceanographic information, whether from in situ platforms or satellite.

## 4.2 Combined upper layer carbon budgets

Taken together with Alkire et al.'s (2012) NCP and carbon export estimates and our adaptation of Briggs et al.'s (2011) depth-resolved carbon fluxes, our productivity and bulk particle estimates provide a remarkably detailed, high-resolution picture of carbon flows over the entire spring bloom. From April 4-17, ML *Chl*, POC, and $O_2$ concentrations changed little, despite large fluctuations in MLD, while $NO_3$ increased slightly during deep-mixing, presumably due to entrainment, but was stable during shallow (<100 m) mixing. Consistent positive 125 m integrated $NPP_{Chl}$ (Fig. 11a) was therefore likely balanced by heterotrophic respiration. From April 18 to May 7, ML shoaling events coincided with several pulses of high net growth in $POC_{cp}$ and *Chl* (Fig. 7), and the close match between $NPP_{Chl}$ and NCP during these periods (Fig. 10) suggests minimal role of grazing in regulating this growth. From May 6-7, all four gliders observed a rapid, aggregation event (Fig. 11b) that triggered a dramatic pulse in carbon export, both from the float patch and the broader (~30 km) glider survey area (Fig. 11a; blue and red lines). This pulse sank through the mesopelagic at ~75 m $d^{-1}$ and was composed primarily of fragile aggregates containing live phytoplankton including *Chaetoceros sp.* resting spores (Martin et al. 2011; Briggs et al. 2011; Rynearson et al. 2013). This aggregate export was the largest loss term of surface POC during the "main bloom", reducing biomass accumulation rate by $\geq$ 50% (Fig. 10). While $SiO_4$ limitation has been proposed as a cause of this rapid sinking event (Bagniewski et al. 2011), this aggregation commenced when $SiO_4$ concentrations were still >2 mmol $m^{-3}$ (Fig. 7a) and five days prior to the ~35% reduction in GOP and $GPP_{cp}$ that we attribute to $SiO_4$ limitation (pink band in Fig. 11b). The exact cause of this rapid aggregation event is unknown, but likely involves a combination of moderately high particle concentration ($POC_{cp}$ > 10 mmol $m^{-3}$), weakening of mixing (which could break fragile aggregates), and production of transparent exopolymer particles (P. Martin et al. 2011; Alkire et al. 2012). The combination of high export and reduced productivity at the end of the diatom bloom (May 12-14) appears to end the ML biomass accumulation. However, we conclude that the subsequent, sharp decline in ML Chl, $POC_{cp}$, and $O_2$ from May 14-15 (Fig. 7) was probably not the result of a dramatic increase in heterotrophic respiration, as implied by the strong negative NCP estimate (Fig. 10) of Alkire et al. (2012). Our conclusion stems from the night-time ML $O_2$ loss rates, which do not increase at all between the bloom peak the bloom decline (see Fig. 4a). Instead, the ML $O_2$ decline appears to be caused by further GPP decreases (Fig. 8b,c), due to continued $SiO_4$ limitation and a decline in *Chl* (Fig. 7b,d), likely enhanced by export of phytoplankton from the shallow ML. The $O_2$ decline (and accelerating *Chl* decline) may have been enhanced by advection of the float relative to the thin surface ML during afternoon profiles (see section 4.1.2), or perhaps an additional, light-dependent process, such as photoinhibition or photorespiration (Spilling et al. 2015), nearly eliminated GPP during this time, but only in the afternoons. After the decline of the diatom bloom, the different productivity estimates again provide a consistent picture, this time of top-down control. GOP and $GPP_{cp}$ again show no sign of nutrient limitation (Fig. 9b,c, red symbols), and $NPP_{Chl}$ is apparently



balanced by heterotrophic respiration. Glider estimates of sinking POC export were low, but higher than early bloom export, despite similar NPP (Fig. 11a) and higher respiration. This result highlights the de-coupling between NCP and export on weekly-monthly timescales in this dynamic system and suggests that biomass and particle size are better predictors of sub-seasonal export dynamics. The changing export efficiencies that we observed (<15% through most of April, to ~ 57% during

the main bloom to ~ 33% in the post-bloom period), provide a complex picture of "the spring bloom", but still agree broadly with the export ratio of 45% calculated by Buesseler and Boyd (2009) in the North Atlantic spring bloom using JGOFS data, among the highest export efficiencies observed in the open ocean. However, unlike Buesseler and Boyd (2009), and in line with the conclusions of Martin et al. (1993), we see significant flux attenuation in the 100 m below the euphotic zone (e.g. 35-48% of flux lost between 60 m and 125 m during the main bloom).

**5 Conclusions**

Our results, placed in the context of previous studies, provide strong support for the diel cycles method as a means to obtain estimates of GOP (from $O_2$) and GPP (from $c_p$) with reasonable accuracy relative to existing methods and enough precision on 3-day timescales to clearly resolve a spring diatom bloom. Because the diel cycles method is well suited for autonomous platforms, it has the potential to greatly increase our coverage of in-situ productivity estimates, providing both direct

knowledge of this critical biological rate and greatly enhanced validation datasets for satellite-derived and modelled productivity. Our results also support the use of short-term $^{14}$C incubations to parameterize simple PvE models for application to autonomous measurements, at least in the absence of strong nutrient limitation. We find high GOP/GPP ratios of 2.1-2.6 through most of the study, suggesting high DOC production and/or a possible moderate under-estimation of GPP by both methods. Finally, combined high-resolution estimates of NPP, particle size and sinking flux during the North

Atlantic spring bloom shows a strong coupling between the three, modulated by a dramatic increase in export efficiency at bloom peak, apparently due to rapid aggregation.

**6 Acknowledgements**

Collection of data for this study was funded by the US National Science Foundation (Grants OCE-0628107 and OCE-0628379) and NASA (Grants NNX-08AL92G and NNX-10AP29H). Analysis and writing was further

funded by a University of Maine Doctoral Research Fellowship, National Science Foundation grant OCE-1420929 and European Research Council grant. The authors would also like to thank Andrew Thomas and Emmanuel Boss for valuable comments and feedback as PhD committee members and the crew and technicians of the *R.V. Knorr* and *R.V. Bjarni Saemundsson* for making this entire study possible.



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



Table 1. Abbreviations used in more than one subsection of the text

| Abbreviation | Description |
| --- | --- |
| $b_{bp}$ | particulate optical backscattering coefficient |
| $Chl$ | Chlorophyll $a$ concentration |
| $c_p$ | particulate optical beam attenuation coefficient |
| $D_{bbp}$ | area-weighted mean particle diameter from optical backscattering |
| DOC | dissolved organic carbon concentration |
| GOP | Gross $O_2$ productivity from $O_2$ diel cycles |
| GPP | Gross primary productivity |
| $GPP_{bbp}$ | GPP from optical backscattering diel cycles |
| $GPP_{Chl}$ | GPP from in situ chlorophyll and light measurements |
| $GPP_{cp}$ | GPP from optical beam attenuation diel cycles |
| JGOFS | Joint Global Ocean Flux Study |
| $K_{PAR}$ | Diffuse attenuation coefficient of PAR |
| ML | Mixed layer |
| MLD | Mixed layer depth |
| NPP | Net primary productivity |
| $NPP_{Chl}$ | NPP from in situ chlorophyll and light measurements |
| PAR | photosynthetically available radiation |
| $P_m$ | Maximum GPP (light saturated) |
| POC | Particulate organic carbon concentration |
| $POC_{bbp}$ | POC from optical backscattering |
| $POC_{cp}$ | POC from optical beam attenuation |
| PP | Primary productivity |
| $R_\Phi$ | Phytoplankton respiration rate |
| $S$ | Salinity |
| $T$ | Temperature |
| $\alpha$ | Initial slope GPP/PAR (light limited) |
| $\varepsilon$ | coefficient representing "sharpness" of NPP vs PAR relationship |





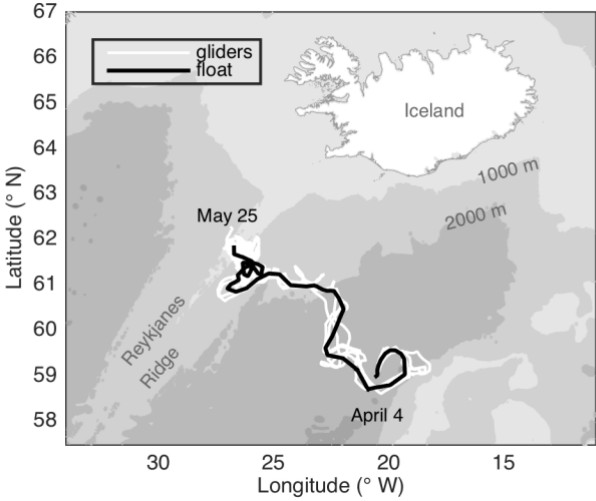

**Fig. 1. Study area with tracks of autonomous Lagrangian mixed-layer float and autonomous Seagliders.**



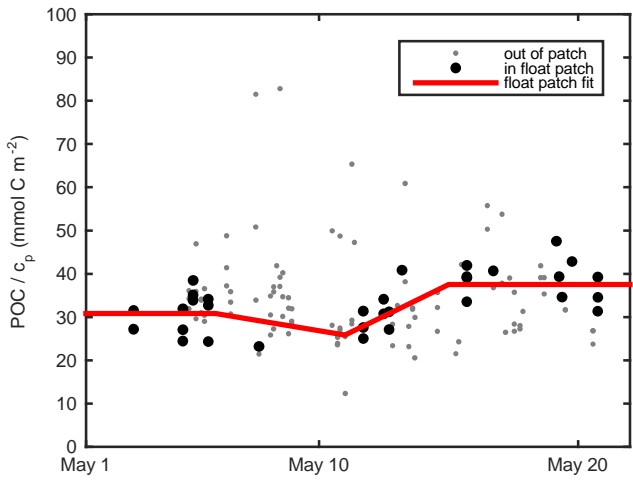

**Fig. 2. POC/$c_p$ from the May cruise in upper 30 m, and fit used to calculate POC$_{cp}$.**





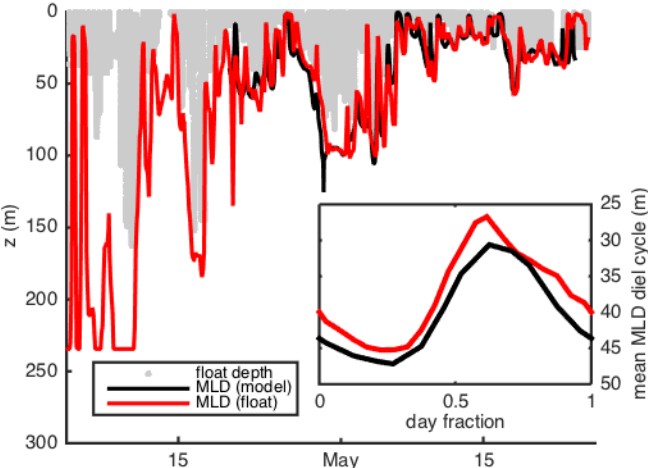

**Fig. 3 Hourly mixed-layer depth estimates calculated directly from float density measurements and from the** Bagniewski et al.
(2011) **data assimilation model, along with the depth of the float in mixed-layer mode. Inset shows mean MLD diel cycle over the
entire duration of the model (April 21-May 24). All MLD estimates use a density threshold of 0.01 kg m$^{-3}$ to better approximate**
5 **active mixing on an hourly timescale.**





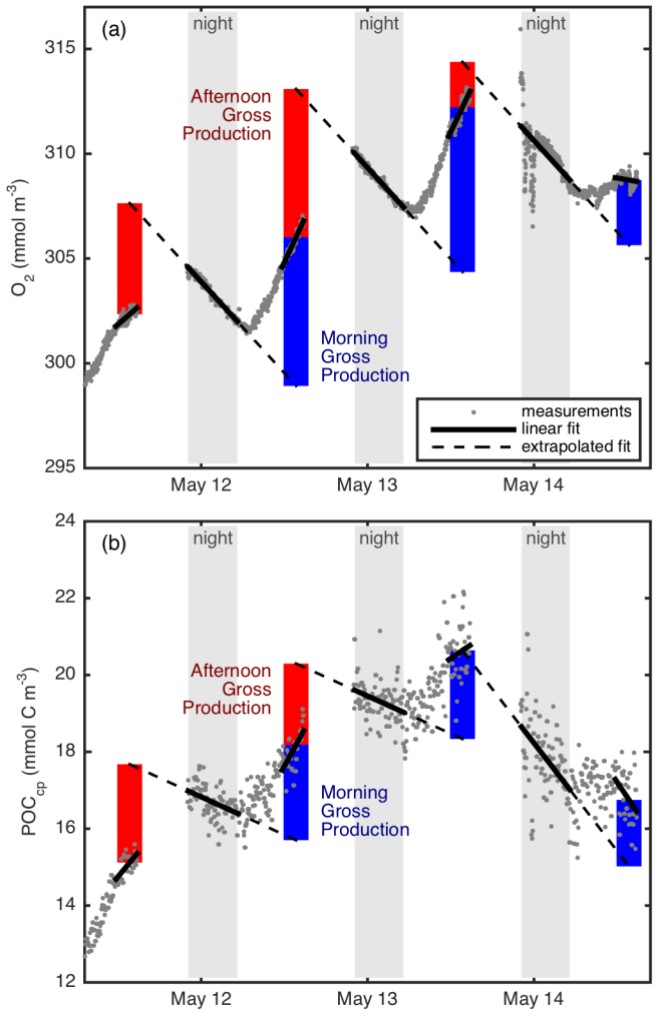

Fig. 4. Calculation of gross production of O₂ (a) and POC_cp (b) in the ML from their diel cycles.





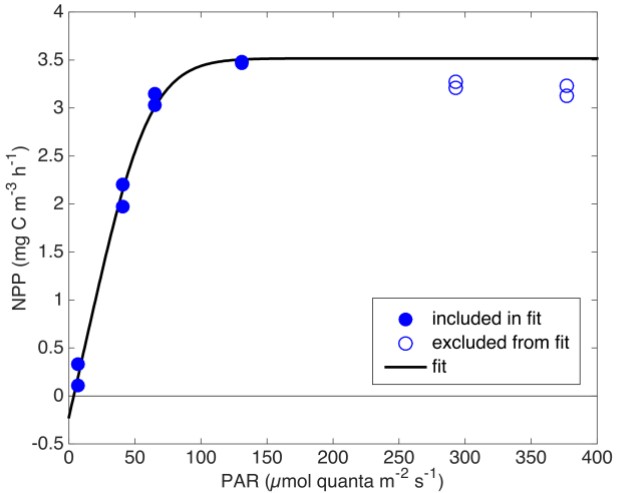

**Fig. 5. Example NPP vs PAR relationship from $^{14}$C incubations, with best fit "PvE" curve.**





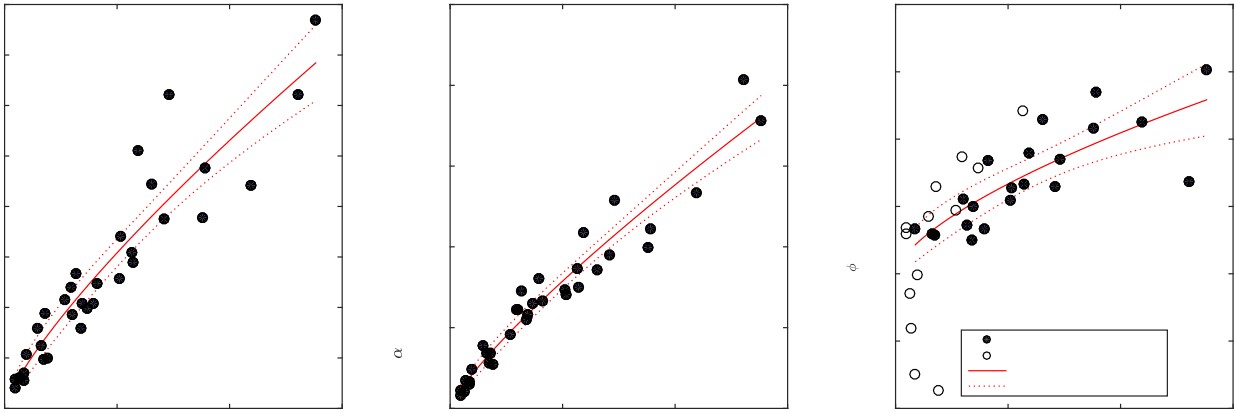

**Fig. 6. Photosynthetic parameters $P_M$ (a), α (b), and $R_\Phi$ (c) vs in situ *Chl* with least squares power law fits and 95% confidence intervals. $R_\Phi$ estimates from April and June cruises are excluded from fit (panel c, open circles).**





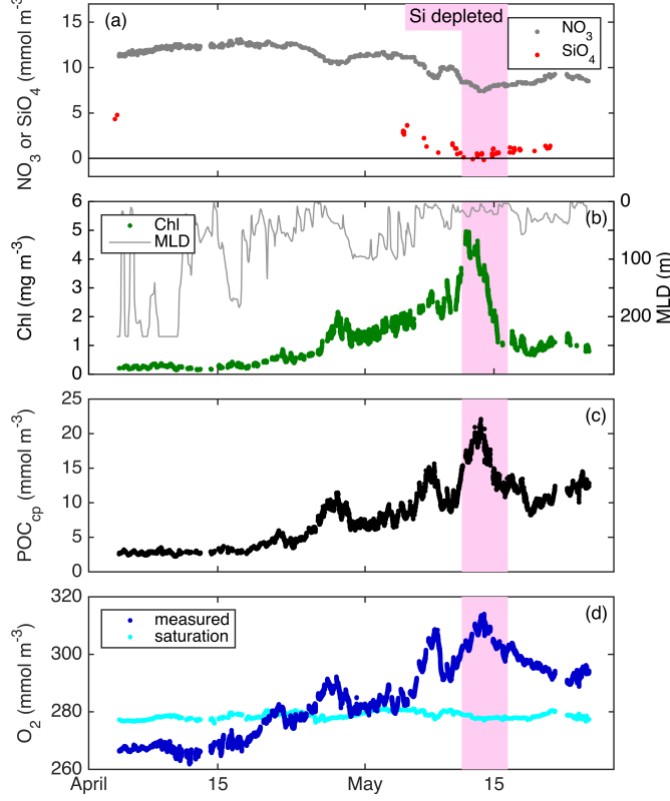

**Fig. 7. Float patch mixed layer timeseries of NO₃ (a), SiO₄ (a), MLD (b), *Chl* (b), POCcp (c), O₂ (d), and the concentration of O₂ saturation (d).**



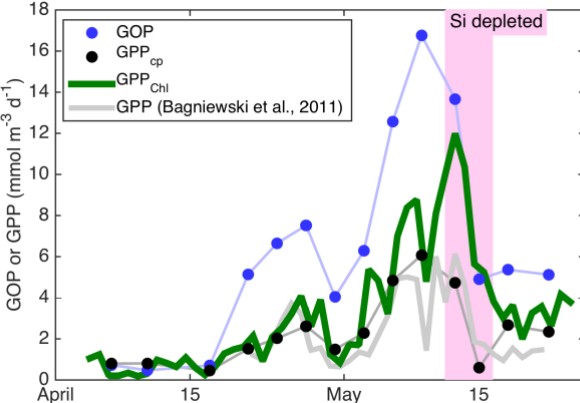

**Fig. 8 Primary productivity estimates within the daily minimum ML. GPP$_{Chl}$, GOP, GPP$_{cp}$ and GPP from Bagniewski et al. (2011). Diel cycles-based estimates are 3-day means; others are daily.**





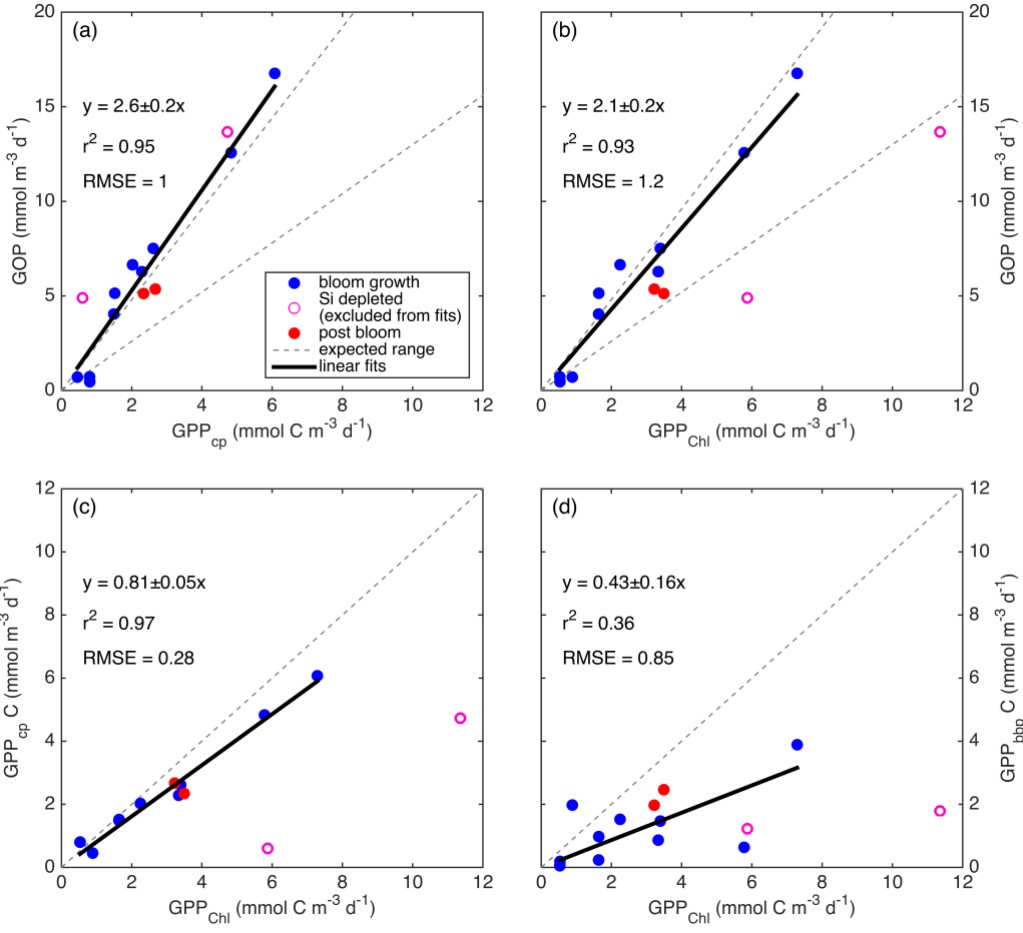

**Fig. 9. Relationships between primary productivity estimates: GOP vs GPP$_{cp}$ (a), GOP vs GPP$_{Chl}$ (b), GPP$_{cp}$ vs GPP$_{Chl}$ (c), and GPP$_{bbp}$ vs GPP$_{Chl}$ (d). Type I linear regressions are forced through the origin and include all data except the SiO$_4$-depleted period (pink circles). Expected range of GOP/GPP (a,b; dashed lines) assumes a photosynthetic quotient between 1-1.45 and 22-40% of fixed carbon released as DOC (see text).**





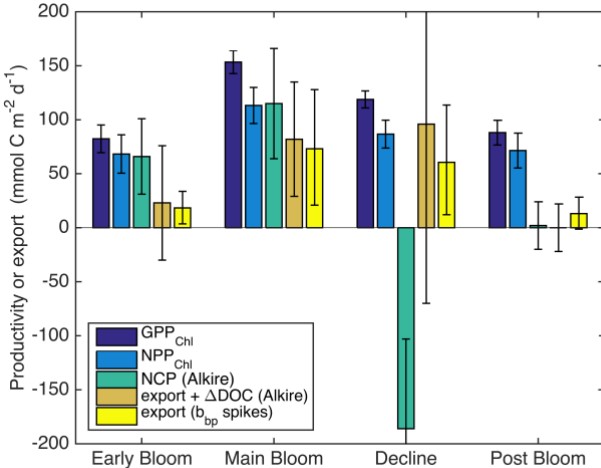

**Fig. 10. Estimates of sources and sinks of organic carbon integrated over the top 60 m: GPP$_{Chl}$ and NPP$_{Chl}$ and sinking particle export (this study), as well as NCP and loss due to the sum of sinking particle export and net DOC production and sinking particle export only** (Alkire et al. 2012)**. Bloom periods follow Alkire et al.** (2012) **and are defined in the text (Section 3.3).**





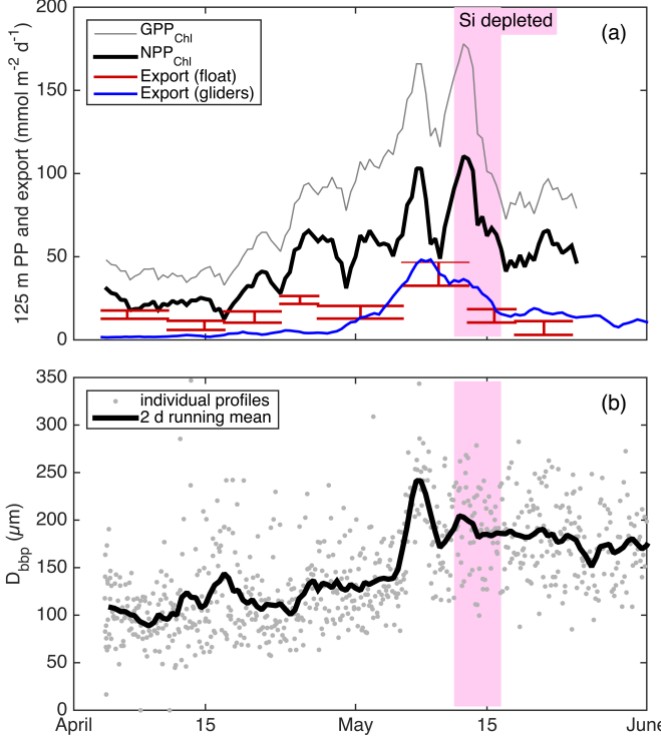

**Fig. 11. (a) Continuous productivity and export from the autonomous float and gliders, to/from the top 125 m over the entire study period. Productivity and glider export are 2-day running means while float export is averaged over longer periods denoted by the width of the bars. Bar height denotes uncertainty bounds. (b) Near surface glider $D_{bbp}$ estimates from 10-50 m.**

