# Peer review of "A multi-method autonomous assessment of primary productivity and export efficiency in the springtime North Atlantic"

_Biogeosciences, 2017_

## Referee Comment (RC1) · Anonymous Referee #1 · 31 Jan 2018

This is a nice exercise, and adds to the growing literature on comparisons of methods for primary production. I have five comments.

1. I'm not sure why the authors chose to cite Cullen et al. (1992). That study doesn't have any actual diel data; any diel relationships were guessed at. For example, if I remember correctly, they simply multiply their change in cp by 10. Also, Cullen et al. (1992) focus on growth rate, not productivity. Growth rate means a normalization to biomass, and therefore a much tougher estimate. I remember reading a recent paper by White et al., published last year (?) in GRL, which would be a better choice.

2. This work is not entirely novel, although I suppose the use of gliders is, and the

incorporation of PvsE estimates. But the same kind of results, with similar good (actually, maybe better) agreement was done in JGOFS' NABE, 20 years before these were done, and reported in Marra (2002) and Marra (2009, Aquat. Microbial Ecol., Fig. 4).

3. It would have been useful to plot the time courses of GPPchl, Chl, and POCcp together. GPPchl looks to be very close to the biomass measures, which means a simple multiplier to get from biomass to productivity. I'm not sure what this means. I would guess they shouldn't be so well matched, and that GPPchl would be expressed earlier in the bloom than Chl or POCcp. That they are well-matched in time, is dubious. In any case, that matchup should be discussed.

4. Bender et al. 1992 is cited incorrectly. The authors list is: Michael Bender, Hugh Ducklow, John Kiddon, John Marra, and John Martin. Makes me think the authors didn't read the paper.

5. In section 2.8.2 there is the phrase: "...incubations were performed at..." Actors "perform," not ocean-going scientists (at least not at sea).

6. I can't find where the authors talk about the environmental limitations in finding their relationships. Will the agreement among the methods that they find only happen when there is a shallowing mixed layer and increasing biomass? Will GPPchl still agree with GPPcp when the mixed layer is deepening, such as during a storm?

---

## Author Comment (AC1) · 28 Feb 2018

We would like to thank Anonymous Referee 1 for this helpful review, which has pointed out several important ways to improve our manuscript. Below are our initial responses (black) to each comment (blue). Final responses with reference to text will be made once we have revised the manuscript.

This is a nice exercise, and adds to the growing literature on comparisons of methods for primary production. I have five comments.

Thank you for your comment. It is also our opinion that this work adds to the broader

literature of PP method comparisons.

Thank you for pointing out the paper by White et al. (2017). This paper is indeed highly relevant, and we apologize to the reviewer and the authors for omitting it. We will need to revise our text slightly regarding the degree of novelty of our beam attenuation results, because this paper contains a more robust quantitative validation of the cp diel cycles method than previous studies that we were aware of.

Thank you for pointing out the work of Marra (2004) from JGOFS. It contains an important quantitative comparison between daytime net POC production from beam attenuation and other estimates of productivity. We will add reference to this work. We disagree, though, with the reviewer's implication that our validation results do not represent a significant advance over this previous work. We see two main advances:

1. We compare two estimates of the same quantity – GPP – and both of our estimates exclude gross DOC production. This allows a more precise validation than comparisons presented in Marra (2002) between differing, but related quantities (daytime net POC production from beam attenuation, 14C assimilation, and net CO2 utilization). The first of these quantities includes loss terms from export and heterotrophic respiration, the second excludes both of these loss terms, and the third method includes

loss from heterotrophic respiration but excludes export and further includes net PIC and DOC production. The White et al. (2017) paper mentioned in the previous comment represents an advance relative to Marra (2002) in this respect, but still compares GP with NPP. 2. Our validation dataset shows not only the mean agreement between methods, but also their correlation over an order of magnitude of productivities, including pre-bloom, diatom bloom, and post-bloom conditions. Analysis of correlation over a high dynamic range adds significant value to a validation exercise. The White et al. (2017) paper mentioned in the previous paper also represents a significant advance in this respect, but our data still have higher dynamic range (factor of 10 vs. factor of 3.5) and much higher maximum GPP (8 vs. 1.75 mmol C/m2/d), adding further value.

We will add these citations and do our best to put or contributions in proper context in the revision. We have also decided to add related work on in situ CO2 and DIC diel cycles to our discussion (e.g. Johnson, 2010 and Merlivat et al., 2015).

3. It would have been useful to plot the time courses of GPPchl, Chl, and POCcp together. GPPchl looks to be very close to the biomass measures, which means a simple multiplier to get from biomass to productivity. I'm not sure what this means. I would guess they shouldn't be so well matched, and that GPPchl would be expressed earlier in the bloom than Chl or POCcp. That they are well-matched in time, is dubious. In any case, that matchup should be discussed.

We agree the precise temporal matchup between changes in GPP and biomass is an interesting topic, and this comparison is well suited to our high-resolution, Lagrangian dataset. We will try to cleanly add the Chl timeseries to Figure 8 of the text to make this matchup clear, but for now, please see Fig. 1 at the end of this comment for the temporal matchup between mixed-layer GPPchl, Chl, and POC estimates. While there is clearly a first-order correlation between GPP and biomass, increases in GPPchl do in fact precede increases in biomass in each rapid growth phase, as the reviewer correctly suggests should be the case. This is due to higher average light in the ML, primarily due to shoaling MLD, but also enhanced by higher surface irradiance. As a third-order
effect, the late April increases in POC slightly precede increases in Chl, perhaps due to reduction in cellular Chl following ML shoaling.

4. Bender et al. 1992 is cited incorrectly. The authors list is: Michael Bender, Hugh Ducklow, John Kiddon, John Marra, and John Martin. Makes me think the authors didn't read the paper.

We apologize for our error in excluding the last two authors and thank the reviewer for catching it. The citation was generated automatically using Mendeley software, which extracts the author list from a PDF, and we did not check the extracted information in enough detail to catch this error. The further suggestion that we did not read the paper, however, is unfounded. The finding that we cite from this paper (GOP/NPP ratio of 2.5) is derived from the ratio of two different numbers in the paper: a GOP/NPP(14h) ratio of 2.0 (Fig. 4 on p1714), and a NPP(24h)/NPP(14h) ratio of 0.8 (on p1712). We are not aware of any way that we could have obtained this number without reading and understanding the relevant parts of the paper.

5. In section 2.8.2 there is the phrase: "...incubations were performed at..." Actors "perform," not ocean-going scientists (at least not at sea).

One of the definitions of the verb "to perform" is "to carry out". If "performed" brings up strange connotations, we'd be happy to replace with the term "carried out".

6. I can't find where the authors talk about the environmental limitations in finding their relationships. Will the agreement among the methods that they find only happen when there is a shallowing mixed layer and increasing biomass? Will GPPchl still agree with GPPcp when the mixed layer is deepening, such as during a storm?

Our validation data span a range of conditions, including periods of ML shoaling, a period of ML deepening at the end of April, increasing biomass, decreasing biomass (Si depletion period), and stable biomass in the post-bloom period. After averaging out some variability due to single episodic events using 3-day means, the methods agree
closely during all of these periods, except the period of Si depletion. We agree with Referee1 that it would be helpful to explicitly discuss this aspect of our findings in the revised text in order to help guide future application of the method.
* * *
[Figure]

**Fig. 1.** Timeseries of mixed-layer GPP and scaled mixed-layer biomass

---

## Referee Comment (RC2) · Anonymous Referee #2 · 6 Mar 2018

This manuscript provides a detailed account of a multi-method assessment of primary production and export efficiency carried out in the North Atlantic between April and June 2008. The research team used an impressive array of autonomous and classical measurement techniques and devoted an important effort to calibrate their instruments. The methodology appears to have been carefully applied and the text is generally well written but difficult to follow in many places (e. g., section 3.2), due to the multiplicity of methods and acronyms (see also comments). The Discussion is thorough and well argued. Overall, this is an interesting manuscript that represents a substantial contribution to marine primary production measurements. Some generally minor comments are given below. Other comments Page 2 Lines 1-7. The term "understanding" appears 5 times in these lines. Perhaps some synonym can also be used. Line 5. "and also of the effects of PP" Page 5 Line 19. Define bbp (It does not appear until line 28). Lines 25-27. I suggest adding some brief background concerning the application of volume backscattering functions and POC estimations. Page 6 Line 16. "a 30 m vertical interval and a 1 day time interval were considered equidistant". Explain more clearly. (The same in page 7, lines 4-5). Page 7. Lines 6-8. Explain more clearly. Line 11. Explain briefly the role of the Bagniewski et al. model, cited in the explanation of Fig. 3 (and later in the text). Line 23 "in-situ KPAR". Is this the KPAR derived from eq. 2? Page 8 Line 15. Define  (greek theta). Line 23 (and following). Air-sea. Page 9 Lines 12-17. Difficult to follow. Explain more clearly. Page 10 Line 10 (eq. 7). It would be helpful to provide some background on the deduction of this empirical model. Page 11 Lines 12.13. Explain more clearly. Perhaps a scheme would help. Line 4. This observation may be valuable for 14C fixation experiments and should be discussed in more detail. Page 12 Lines 5-10. Figure 8 does not have indcations a, b, c . . . Line 8. Where is GPPbbp in Fig. 8? Line 11. "both GOP/GPPChl and GPPcp/GPPChl were substantially lower" Lower than what? Line 24. Eliminate "depth-integrated" (repeated later). Page 13 Lines 1-2. It would be helpful to indicate that this "apparent community respiration" refers to the negative NCP. Line 22 (and page 14, line 2). Indicate that the slope is given in Fig. 9. Page 14. Line 16. Revise sentence. Page 15 Line 1. Eliminate the first Ân the Âż. Line 32. "advection of the float realtive to ML". Explain more clearly. Page 19 Line s 9-10. Where can we see the "flux attenuation in the 100 m below the euphotic zone"?.

---

## Author Comment (AC2) · 20 Mar 2018

We would like to thank Anonymous Referee 2 for this thorough and very helpful review, which has pointed out a number of minor errors in the text as well as areas in need of clarification. We agree with essentially all of this referee's comments, and incorporation of this feedback should substantially improve the clarity (and usefulness) of this manuscript. Below are our individual responses (black) to each comment (blue).

This manuscript provides a detailed account of a multi-method assessment of primary production and export efficiency carried out in the North Atlantic between April and June 2008. The research team used an impressive array of autonomous and classical

measurement techniques and devoted an important effort to calibrate their instruments. The methodology appears to have been carefully applied and the text is generally well written but difficult to follow in many places (e. g., section 3.2), due to the multiplicity of methods and acronyms (see also comments). The Discussion is thorough and well argued. Overall, this is an interesting manuscript that represents a substantial contribution to marine primary production measurements. Some generally minor comments are given below.

Thank you for your kind words and your very careful review. We are pleased that you find our work to be a substantial contribution to the primary productivity literature, and we appreciate your work to improve this contribution.

Other comments

Page 2 Lines 1-7. The term "understanding" appears 5 times in these lines. Perhaps some synonym can also be used.

We agree that this wording should be changed.

Line 5. "and also of the effects of PP"

We agree that this wording should be changed.

Page 5 Line 19. Define bbp (It does not appear until line 28).

We agree.

Lines 25-27. I suggest adding some brief background concerning the application of volume backscattering functions and POC estimations.

Good suggestion. We can add a single sentence to the beginning of the previous section (POC from beam attenuation) citing previous work linking light scattering measurements (including beam attenuation and backscattering) to POC in open ocean waters (low inorganic sediment load).

Page 6 Line 16. "a 30 m vertical interval and a 1 day time interval were considered equidistant". Explain more clearly. (The same in Page 7, lines 4-5).

We used triangulation-based 2-D linear interpolation (Matlab function griddata). For the purposes of this interpolation, the distance between points was calculated as $[(z1/30-z2/30)^2 + (t1 - t2)^2]^0.5, where z is depth in meters and t is time in days. This favors interpolation in time when time gaps between measurements (i$

Page 7. Lines 6-8. Explain more clearly.

When the float is actively profiling (not following the vertical motion of the water), it could entrain water, over-estimating MLD during downward profiles and under-estimating MLD during upward profiles. However, the profile data are critical to the MLD calculation and cannot be discarded. Therefore, the MLD is calculated twice, once using only downward profiles and once using only upward profiles. Note that this method also smooths out effects of internal waves, which can make the depth of an isopycnal in a single profile (up or down) unrepresentative of the mean isopycnal depth. Profiles were distinguished from Lagrangian or near-Lagrangian motion using a vertical velocity threshold of 1 m min-1. We can slightly expand our explanation in the text to make this method and its motivation clearer.

Line 11. Explain briefly the role of the Bagniewski et al. model, cited in the explanation of Fig. 3 (and later in the text).

The MLD determination described in this section does not utilize the Bagniewski et al. model. The temperature and salinity fields of the model are strongly constrained by the daily float profiles, but the diel mixing dynamics are slightly different, so MLD was calculated separately using the model output. This calculation used nearly the same method as described here. For each model timestep, MLD was the shallowest depth where the potential density anomaly exceeded the minimum potential density anomaly by $\geq$ 0.01 kg m$^-3$. However, the first three steps, smoothing, binning, and interpolating, were not needed, because the model output is already

Line 23 "in-situ KPAR". Is this the KPAR derived from eq. 2?

Yes. We should replace the term KPAR in this sentence with the more precise term KPAR(measured)

Page 8 Line 15. Define ï ËŻAs (greek theta).

Agreed. Thank you for catching this.

Line 23 (and following). Air-sea.

We agree that a hyphen should be added.

Page 9 ′ Lines 12-17. Difficult to follow. Explain more clearly.

We can rework this section if it is not clear and run it by other colleagues for clarity.

Page 10 Line 10 (eq. 7). It would be helpful to provide some background on the deduction of this empirical model.

The equation is based on a conceptual model that there is a limiting step in photosynthesis that becomes saturated when it receives too much energy at once. The energy comes in packets and if too many packets arrive during the same period of time, then some energy is wasted. The epsilon parameter denotes how many packets can be received at once without being wasted. It represents a sort of energy "buffer" at the rate-limiting step. It was introduced because empirical models without a buffer don't seem to fit our observations. We can add text to such effect.

Page 11 Lines 12.13. Explain more clearly. Perhaps a scheme would help.

Separation into large and small particles follows the method of Briggs et al. (2011). We will try make this text clearer and also state that a visual schematic is available in Briggs et al. (2011) if further clarification is needed.

Line 4. This observation may be valuable for 14C fixation experiments and should be discussed in more detail.

We assume this comment refers to line 4 of page 10 (our conclusion, based on in situ dO2/dt, that bottle photoinhibition is not representative of most field conditions). We agree that it would be useful to briefly mention and discuss this finding in the discussion section.

Page 12 Lines 5-10. Figure 8 does not have indcations a, b, c . . .

Thank you for catching this error. This text referred to a previous version of figure 8.

Line 8. Where is GPPbbp in Fig. 8?

Thank you for catching this error. This text referred to a previous version of figure 8.

Line 11. "both GOP/GPPChl and GPPcp/GPPChl were substantially lower" Lower than what?

Lower than during the bloom growth phase. We agree that this sentence should be clarified.

Line 24. Eliminate "depth-integrated" (repeated later).

Thank you for catching this error.

Page 13 Lines 1-2. It would be helpful to indicate that this "apparent community respiration" refers to the negative NCP.

We can add "apparent community respiration (difference between GPPChl and NCP)"

Line 22 (and Page 14, line 2). Indicate that the slope is given in Fig. 9.

We agree that this clarification would be helpful.

Page 14. Line 16. Revise sentence.

Thank you for noticing this error. The sentence is missing an "and". It should read: "This conclusion agrees with the coupled physical-biological model of Bagniewski et al. (2011), which assimilated float biogeochemical measurements AND achieved optimal

fit when diatom GPP was limited by SiO4 with a half-saturation constant of 1 $\mu$mol m-3."

Page 15 Line 1. Eliminate the first "the".

Agreed. Thank you.

Line 32. "advection of the float realtive to ML". Explain more clearly. ËŹ

We are referring to horizontal advection of the float relative to the mixed layer during the hours that the float is below the mixed layer.

Page 19 Line s 9-10. Where can we see the "flux attenuation in the 100 m below the euphotic zone"?.

This statement refers a comparison between export estimates from 60 m (Fig. 10) and export at 125 m (Fig. 11a) during the main bloom. We agree that this should be clarified in the text.

---

## Author Response (AR1)

Dear Associate Editor,

We very much appreciate the feedback from the two reviewers, and we feel that incorporation of this feedback has substantially improved and refined this manuscript. We have responded to each comment below, and we have edited parts the text and one figure (referenced below) in response to each referee suggestion as well. The line number references refer to the non-marked-up version of the manuscript that is uploaded. The number references in the marked-up version appended at the end of this document differ due to the in-line inclusion of deleted text. Note that most of our responses remain unchanged from our initial replies already published in the discussion. In addition to responding to the referees, we have taken another look through the manuscript and caught and corrected a few remaining minor errors (visible in the markup). We hope that we have appropriately and fully answered all of the referee's concerns to your (and their) satisfaction.

Sincerely,
Nathan Briggs

Responses to Referee #1:

We would like to thank Anonymous Referee #1 for this helpful review, which has led to several important improvements to our manuscript. Below are our responses (black) to each comment (blue).

This is a nice exercise, and adds to the growing literature on comparisons of methods for primary production. I have five comments.

Thank you for your comment. It is also our opinion that this work adds to the broader literature of PP method comparisons.

1. I'm not sure why the authors chose to cite Cullen et al. (1992). That study doesn't have any actual diel data; any diel relationships were guessed at. For example, if I remember correctly, they simply multiply their change in cp by 10. Also, Cullen et al. (1992) focus on growth rate, not productivity. Growth rate means a normalization to biomass, and therefore a much tougher estimate. I remember reading a recent paper by White et al., published last year (?) in GRL, which would be a better choice.

Thank you for pointing out the paper by White et al. (2017). This paper is indeed highly relevant, and we apologize to the reviewer and the authors for omitting it. We have revised part of our discussion to include these results (p15 lines 27-30)

2. This work is not entirely novel, although I suppose the use of gliders is, and the incorporation of PvsE estimates. But the same kind of results, with similar good (actually, maybe better) agreement was done in JGOFS' NABE, 20 years before these were done, and reported in Marra (2002) and Marra (2009, Aquat. Microbial Ecol., Fig. 4).

Thank you for pointing out the work of Marra (2002) from JGOFS. It contains an important quantitative comparison between daytime net POC production from beam attenuation and other estimates of productivity. We have added reference to this work. We disagree, though, with the reviewer's implication that our methods and validation dataset do not represent a significant advance over this previous work. We see two main advances:

1. We compare two estimates of the same quantity: GPP, and both estimates exclude gross DOC production. This allows a more precise validation than comparisons presented in Marra (2002) between differing, but related quantities: daytime net POC production from beam

attenuation, 14C assimilation, and net CO2 utilization. The first of these quantities includes loss terms from export and heterotrophic respiration, the second excludes both of these loss terms, and the third method includes loss from heterotrophic respiration but excludes export and further includes net PIC and DOC production. The White et al. (2017) paper mentioned in the previous paper represents an advance relative to Marra (2002) in this respect, but still compares GP with NPP.

2. Our validation dataset shows not only the mean agreement between methods, but also their correlation over an order of magnitude of productivities, including pre-bloom, diatom bloom, and post-bloom conditions. Analysis of correlation over a high dynamic range adds significant value to a validation exercise. The White et al. (2017) paper mentioned in the previous comment also represents a significant advance in this respect, but our data still have higher dynamic range (factor of 10 vs. factor of ~3.5) and much higher maximum GPP (8 vs. 1.75 mmol C/m2/d), adding further value.

In addition to adding the suggested citations, we have gone through the literature again and added some further references to our discussion, including very recent work on bbp diel cycles (Poulin et al., 2018) and work on in situ CO2 diel cycles (e.g. Johnson, 2010 and Merlivat et al., 2015).

3. It would have been useful to plot the time courses of GPPchl, Chl, and POCcp together. GPPchl looks to be very close to the biomass measures, which means a simple multiplier to get from biomass to productivity. I'm not sure what this means. I would guess they shouldn't be so well matched, and that GPPchl would be expressed earlier in the bloom than Chl or POCcp. That they are well-matched in time, is dubious. In any case, that matchup should be discussed.

We agree that it is interesting to show the precise temporal matchup between GPP and biomass, a comparison well suited to our autonomous, Lagrangian dataset. We have added the Chl timeseries to Fig. 8. While there is clearly a first-order correlation between GPP and biomass, increases in $GPP_{chl}$ do in fact precede increases in biomass in each rapid growth phase, as the reviewer correctly suggests should be the case. This is due to higher average light in the ML, primarily due to shoaling MLD, but also enhanced by higher surface irradiance. As a third-order effect, not shown in the text, but shown in our initial online reply to this comment, increases in POC slightly precede increases in Chl, perhaps due to reduction in cellular Chl following ML shoaling.

4. Bender et al. 1992 is cited incorrectly. The authors list is: Michael Bender, Hugh Ducklow, John Kiddon, John Marra, and John Martin. Makes me think the authors didn't read the paper.

We apologize for our error in excluding the last two authors and thank the reviewer for catching it. The citation was generated automatically using Mendeley software, which extracts the author list from a PDF, and we did not check the extracted information in enough detail to catch this error. The further suggestion that we did not read the paper, however, is unfounded. The finding that we cite from this paper (GOP/NPP ratio of 2.5) is derived from the ratio of two different numbers in the paper: a GOP/NPP(14h) ratio of 2.0 (Fig. 4 on p1714), and a NPP(24h)/NPP(14h) ratio of 0.8 (on p1712). We are not aware of any way that we could have obtained this number without reading and understanding the relevant parts of the paper.

5. In section 2.8.2 there is the phrase: "...incubations were performed at..." Actors "perform," not ocean-going scientists (at least not at sea).

We have replaced this instance of the verb "to perform" with the term "to carry out".

6. I can't find where the authors talk about the environmental limitations in finding their relationships. Will the agreement among the methods that they find only happen when there is a shallowing mixed layer and increasing biomass? Will GPPchl still agree with GPPcp when the mixed layer is deepening, such as during a storm?

Our validation data span a range of conditions, including periods of ML shoaling, a period of ML deepening at the end of April, increasing biomass, decreasing biomass (Si depletion period), and stable biomass in the post-bloom period. After averaging out some variability due to single episodic events using 3-day means, the methods agree closely during all of these periods, except the period of Si depletion. We add some text in the conclusions (p19, lines 28-30) to emphasize the evidence so far for the broader applicability of diel cycles methods, both from our study and from other studies in different ocean basins.

Responses to Referee #2:

We would like to thank Anonymous Referee #2 for this thorough and very helpful review, which has pointed out a number of minor errors in the text as well as areas in need of clarification. We agree with essentially all of this referee's comments, and incorporation of this feedback has substantially improved the clarity (and usefulness) of this manuscript. Below are our individual responses (black) to each comment (blue) with citations to the changed text.

This manuscript provides a detailed account of a multi-method assessment of primary production and export efficiency carried out in the North Atlantic between April and June 2008. The research team used an impressive array of autonomous and classical measurement techniques and devoted an important effort to calibrate their instruments. The methodology appears to have been carefully applied and the text is generally well written but difficult to follow in many places (e. g., section 3.2), due to the multiplicity of methods and acronyms (see also comments). The Discussion is thorough and well argued. Overall, this is an interesting manuscript that represents a substantial contribution to marine primary production measurements. Some generally minor comments are given below.

Thank you for your kind words and your very careful review. We are pleased that you find our work to be a substantial contribution to the primary productivity literature, and we appreciate your work to improve this contribution.

Other comments

Page 2 Lines 1-7. The term "understanding" appears 5 times in these lines. Perhaps some synonym can also be used.

Thank you, we have changed the wording, using "understanding" only twice.

Line 5. "and also of the effects of PP"

We have changed the phrase to "the drivers of PP and its effects on ecosystems".

Page 5 Line 19. Define bbp (It does not appear until line 28).

bbp is now defined in section 2.1 (page 3, line 19).

Lines 25-27. I suggest adding some brief background concerning the application of volume backscattering functions and POC estimations.

Good suggestion. We have added two sentences to the beginning of the previous section (POC from beam attenuation) for background: "Previous work has shown that measurements of light scattering by particles, including beam attenuation $c_p$ and particulate backscattering $b_{bp}$ correlate strongly with POC in the open ocean (Cetinic et al. 2012 and references therein). Calibration of our $c_p$ and $b_{bp}$ measurements and conversion to POC estimates are described in the next two subsections."

Page 6 Line 16. "a 30 m vertical interval and a 1 day time interval were considered equidistant". Explain more clearly. (The same in Page 7, lines 4-5).

We used triangulation-based 2-D linear interpolation (Matlab function griddata). For the purposes of this interpolation, the distance between points was calculated as $[(z1/30-z2/30)^2 +(t1-t2)^2 ]^{0.5}$ ,

where z is depth in meters and t is time in days. This favors interpolation in time when time gaps between measurements (in days) are less than 1/30 of vertical spatial gaps in measurements (in meters), and vice versa. We have added text to clarify (p6, new lines 18-19).

When the float is actively profiling (not following the vertical motion of the water), it could entrain water, over-estimating MLD during downward profiles and under-estimating MLD during upward profiles. However, the profile data are critical to the MLD calculation and cannot be discarded. Therefore, the MLD is calculated twice, once using only downward profiles and once using only upward profiles. Note that this method also smooths out effects of internal waves, which can make the depth of an isopycnal in a single profile (up or down) unrepresentative of the mean isopycnal depth. Profiles were distinguished from Lagrangian or near-Lagrangian motion using a vertical velocity threshold of 1 m min$^{-1}$.

We can have slightly expanded our explanation in the text to make this method and its motivation clearer (p7, lines 10-11).

Line 11. Explain briefly the role of the Bagniewski et al. model, cited in the explanation of Fig. 3 (and later in the text).

The MLD determination described in this section does not utilize the Bagneiwski et al. model. The temperature and salinity fields of the model are strongly constrained by the daily float profiles, but the diel mixing dynamics are slightly different, so MLD was calculated separately using the model output. This calculation used nearly the same method as described here. For each model timestep, MLD was the shallowest depth where the potential density anomaly exceeded the minimum potential density anomaly by $\geq 0.01$ kg/m^3. We have added text to clarify (p7, lines 13-15)

Line 23 "in-situ KPAR". Is this the KPAR derived from eq. 2?

Yes. We replaced the term KPAR in this sentence with the more precise term KPAR(measured)

Page 8 Line 15. Define ï¸ As (greek theta).

Agreed. Thank you for catching this. We have added the definition (solar zenith angle).

Line 23 (and following). Air-sea.

We have added the hyphen.

Page 9 ´ Lines 12-17. Difficult to follow. Explain more clearly.

We have added text to this section for clarity on p9, lines 18 and 22-24.

Page 10 Line 10 (eq. 7). It would be helpful to provide some background on the deduction of this empirical model.

The equation is based on a conceptual model that there is a limiting step in photosynthesis that becomes saturated when it receives too much energy at once. The energy comes in packets and if too many packets arrive during the same period of time, then some energy is wasted. The epsilon parameter denotes how many packets can be received at once without being wasted. It represents a sort of energy "buffer" at the rate-limiting step. It was introduced because empirical models without a buffer don't seem to fit our observations. We have added text to such effect on lines 19-21.

Page 11 Lines 12.13. Explain more clearly. Perhaps a scheme would help.

Separation into large and small particles follows the method of Briggs et al. (2011). We have added text to clarify.

Line 4. This observation may be valuable for 14C fixation experiments and should be discussed in more detail.

We assume this comment refers to line 4 of page 10 (our conclusion, based on in situ dO2/dt, that bottle photoinhibition is not representative of most field conditions). We agree that this is an interesting result with wider relevance and added discussion on p15 lines 9-13.

Page 12 Lines 5-10. Figure 8 does not have indcations a, b, c . . .

Thank you for catching this error. This text referred to a previous version of figure 8.

Line 8. Where is GPPbbp in Fig. 8?

Thank you for catching this error. This text referred to a previous version of figure 8.

Line 11. "both GOP/GPPChl and GPPcp/GPPChl were substantially lower" Lower than what?

Lower than during the bloom growth phase. We have added this clarification.

Line 24. Eliminate "depth-integrated" (repeated later).

Thank you for catching this error. We have corrected it as suggested.

Page 13 Lines 1-2. It would be helpful to indicate that this "apparent community respiration" refers to the negative NCP.

We have added clarification in parentheses: "apparent community respiration (difference between GPPChl and NCP)"

Line 22 (and Page 14, line 2). Indicate that the slope is given in Fig. 9.

Done.

Page 14. Line 16. Revise sentence.

Thank you for noticing this error. The sentence was missing an "and". It now reads:
"This conclusion agrees with the coupled physical-biological model of Bagniewski et al. (2011), which assimilated float biogeochemical measurements AND achieved optimal fit when diatom GPP was limited by SiO4 with a half-saturation constant of 1 μmol m^{-3}."

Page 15 Line 1. Eliminate the first "the".

Fixed. Thank you.

Line 32. "advection of the float realtive to ML". Explain more clearly. ˙

We are referring to horizontal advection of the float relative to the mixed layer during the hours that the float is below the mixed layer. New text (p16, lines 12-14) reads as follows: "One plausible explanation is horizontal advection of the float relative to the ML during its afternoon profile, causing it to resurface in water with lower biomass."

Page 19 Line s 9-10. Where can we see the "flux attenuation in the 100 m below the euphotic zone"?.

This statement refers a comparison between export estimates from 60 m (Fig. 10) and export at 125 m (Fig. 11a) during the main bloom. We have clarified this in the text (p19, lines 23-24).

[revised manuscript text omitted]